# RhoGAP19D inhibits Cdc42 laterally to control epithelial cell shape and prevent invasion

Weronika Fic[1,2], Rebecca Bastock[1,2], Francesco Raimondi[3], Erinn Los[1,2], Yoshiko Inoue[1,4], Jennifer L. Gallop[1,4], Robert B. Russell[3], and Daniel St Johnston[1,2]

**Cdc42-GTP is required for apical domain formation in epithelial cells, where it recruits and activates the Par-6–aPKC polarity complex, but how the activity of Cdc42 itself is restricted apically is unclear. We used sequence analysis and 3D structural modeling to determine which *Drosophila* GTPase-activating proteins (GAPs) are likely to interact with Cdc42 and identified RhoGAP19D as the only high-probability Cdc42GAP required for polarity in the follicular epithelium. RhoGAP19D is recruited by α-catenin to lateral E-cadherin adhesion complexes, resulting in exclusion of active Cdc42 from the lateral domain. *rhogap19d* mutants therefore lead to lateral Cdc42 activity, which expands the apical domain through increased Par-6/aPKC activity and stimulates lateral contractility through the myosin light chain kinase, Genghis khan (MRCK). This causes buckling of the epithelium and invasion into the adjacent tissue, a phenotype resembling that of precancerous breast lesions. Thus, RhoGAP19D couples lateral cadherin adhesion to the apical localization of active Cdc42, thereby suppressing epithelial invasion.**

## Introduction

The form and function of epithelial cells depends on their polarization into distinct apical, lateral, and basal domains by conserved polarity factors (Rodriguez-Boulan and Macara, 2014; St Johnston and Ahringer, 2010). This polarity is then maintained by mutual antagonism between apical polarity factors such as atypical PKC (aPKC) and lateral factors such as Lethal (2) giant larvae (Lgl) and Par-1. While many aspects of the polarity machinery are now well understood, it is still unclear how the apical domain is initiated and what role cell division control protein 42 (Cdc42) plays in this process.

Cdc42 was identified for its role in establishing polarity in budding yeast, where it targets cell growth to the bud tip by polarizing the actin cytoskeleton and exocytosis toward a single site (Chiou et al., 2017). It has subsequently been found to function in the establishment of cell polarity in multiple contexts. For example, Cdc42 recruits and activates the anterior PAR complex to polarize the anterior–posterior axis in the *Caenorhabditis elegans* zygote and the apical–basal axis during the asymmetric divisions of *Drosophila* neural stem cells (Gotta et al., 2001; Kay and Hunter, 2001; Atwood et al., 2007; Rodriguez et al., 2017).

Cdc42 also plays an essential role in the apical–basal polarization of epithelial cells, where it is required for apical domain formation (Genova et al., 2000; Hutterer et al., 2004; Jaffe et al., 2008; Bray et al., 2011; Fletcher et al., 2012). Cdc42 is active when bound to GTP, which changes its conformation to allow it to bind downstream effector proteins that control the cytoskeleton and membrane trafficking. An important Cdc42 effector in epithelial cells is the Par-6–aPKC complex. Par-6 binds directly to the switch 1 region of Cdc42 GTP through its semi-CRIB domain (Cdc42 and Rac interactive binding; Lin et al., 2000; Joberty et al., 2000; Qiu et al., 2000; Yamanaka et al., 2001). This induces a change in the conformation of Par-6 that allows it to bind to the C-terminus of another key apical polarity factor, the transmembrane protein Crumbs, which triggers the activation of aPKC's kinase activity (Peterson et al., 2004; Whitney et al., 2016; Dong et al., 2020). As a result, active aPKC is anchored to the apical membrane, where it phosphorylates and excludes lateral factors, such as Lgl, Par-1, and Bazooka (Baz; Betschinger et al., 2003; Hurov et al., 2004; Suzuki et al., 2004; Nagai-Tamai et al., 2002; Morais-de-Sá et al., 2010). In addition to this direct role in apical–basal polarity, Cdc42 also regulates the organization and activity of the apical cytoskeleton through effectors such as neuronal Wiskott-Aldrich syndrome protein (N-WASP), which promotes actin polymerization, and myotonic dystrophy kinase-related Cdc42-binding kinase (MRCK; Genghis khan [*Gek*] in *Drosophila*), which phosphorylates the myosin regulatory light chain to activate contractility (Padrick and Rosen, 2010; Zihni et al., 2017).

[1]Gurdon Institute, University of Cambridge, Cambridge, UK; [2]Department of Genetics, University of Cambridge, Cambridge, UK; [3]BioQuant and Biochemie Zentrum Heidelberg, Heidelberg University, Heidelberg, Germany; [4]Department of Biochemistry, University of Cambridge, Cambridge, UK.

Correspondence to Daniel St Johnston: d.stjohnston@gurdon.cam.ac.uk; R. Bastock's present address is Sheffield Diagnostic Genetics Service, Sheffield Children's NHS Trust, Western Bank, Sheffield, UK; F. Raimondi's present address is Bio@SNS, Scuola Normale Superiore, Pisa, Italy; Y. Inoue's present address is Research Organization of Information and Systems, Minato-ku, Tokyo, Japan.



This crucial role of active Cdc42 in specifying the apical domain raises the question of how Cdc42-GTP itself is localized apically. In principle, this could involve activation by Cdc42 guanine nucleotide exchange factors (Cdc42GEFs) that are themselves apical or lateral inactivation by Cdc42GAPs. The Cdc42GEFs Tuba, intersectin 2, and Dbl3 have been implicated in activating Cdc42 in mammalian epithelia (Otani et al., 2006; Qin et al., 2010; Rodriguez-Fraticelli et al., 2010; Oda et al., 2014; Zihni et al., 2014). Only Dbl3 localizes apical to tight junctions, however, as Tuba is cytoplasmic and enriched at tricellular junctions and intersectin 2 localizes to centrosomes. Thus, GEF activity may not be exclusively apical, suggesting that it is more important to inhibit Cdc42 laterally. Although nothing is known about the role of GAPs in restricting Cdc42 activity to the apical domain of epithelial cells, this mechanism plays an instructive role in establishing radial polarity in the blastomeres of the early *C. elegans* embryo. In this system, the Cdc42GAP PAC-1 is recruited by the cadherin adhesion complex to sites of cell–cell contact, thereby restricting active Cdc42 and its effector the Par-6–aPKC complex to the contact-free surface (Anderson et al., 2008; Klompstra et al., 2015).

Here we analyzed the roles of Cdc42GAPs in epithelial polarity using the follicle cells that surround developing *Drosophila* egg chambers as a model system (Bastock and St Johnston, 2008). By generating mutants in a number of candidate Cdc42GAPs, we identified the Pac-1 orthologue, RhoGAP19D, as the GAP that restricts active Cdc42 to the apical domain. In the absence of RhoGAP19D, lateral Cdc42 activity leads to an expansion of the apical domain and a high frequency of epithelial invasion into the germline tissue, a phenotype that mimics the early steps of carcinoma formation.

## Results

To confirm that Cdc42 regulates apical domain formation in *Drosophila* epithelia, we generated homozygous mutant clones of *cdc42²*, a null allele, in the follicular epithelium that surrounds developing egg chambers (Fig. 1 A). Mutant cells lose their cuboidal shape, leading to gaps and multilayering in the epithelium, and fail to localize GFP-aPKC apically, indicating that Cdc42 is required for polarity in the follicle cell layer.

There are 22 Rho-GTPase–activating proteins in the *Drosophila* genome (Table 1), but in most cases, it is unclear whether they regulate Rho, Rac, or Cdc42. We therefore predicted the tendency for each of the *Drosophila* GAPs to interact with Cdc42 using InterPReTS (Aloy and Russell, 2002). This uses a known structure of a protein complex (in this case, the structure of the human CDC42–Rho-GTPase–activating protein 1 [ARHGAP1] complex; Nassar et al., 1998) as a template to predict whether homologous proteins (in this case, other *Drosophila* GAPs and Cdc42) would be able to interact in the same way. The fit of each sequence pair on the structure is assessed via statistical potentials that score the compatibility of each amino acid pair at the (e.g., GAP–Cdc42) interface. The rank for each *Drosophila* GAP according to its likelihood of interacting with *Drosophila* Cdc42 is shown in Table 2.

Interestingly, the *Drosophila* ARHGAP1 orthologue, RhoGAP68F, was only second in the ranking behind another known Cdc42GAP, the PAC-1 orthologue, RhoGAP19D. Inspection of the InterPreTS results in detail shows that several key conserved positions mediating interactions in the structure of the human CDC42–ARHGAP1 complex are conserved in RhoGAP19D (and many other) *Drosophila* GAPs, in addition to three positions that appear to make the interaction stronger (Fig. 1, B and C). Specifically, an Arg in ARHGAP1 is replaced by an Ile (1237 in RhoGAP19D), making for a better interaction with Ala13 in Cdc42; a Val is replaced by a Ser (1275), making a more favorable interaction with Glu62; and a Thr is replaced by a Lys (1241), possibly making an additional salt bridge with Glu95 and additional interactions with Asn92.

To test whether any of these putative Cdc42 GAPs play a role in epithelial polarity, we generated null mutants in the seven highest-ranked GAPs using CRISPR-mediated mutagenesis (Table S1) and examined their phenotypes in the follicular epithelium that surrounds developing egg chambers. Mutants in RhoGAP92B are lethal, and we therefore used the flippase (Flp)/flippase recognition target (FRT) system to generate homozygous mutant follicle cell clones, whereas mutants in the other GAPs are homozygous viable or semiviable, allowing us to analyze the follicle cell phenotype in homozygous mutant females. Null mutants in *RhoGAP68F*, *CdGAPr*, *RhoGAP92B*, *RhoGAP82C*, *conundrum* (Conu; Neisch et al., 2013), and *RhoGAP93B* cause no discernible changes in follicle cell shape or polarity, as shown by the localization of aPKC apically and Lgl laterally (Fig. S1). By contrast, 40% of homozygous mutant *rhogap19d* egg chambers show invasions of regions of the follicular epithelium into the overlying germline cyst (Fig. 2, A–C). This phenotype is not a consequence of overproliferation of the mutant follicle cells, because mutant egg chambers contain the same number of follicle cells as wild-type egg chambers, and the same proportion of homozygous mutant cells and wild-type cells are in mitosis at stages 4 and 5 (Fig. 2, D–G). Introducing endogenously tagged E-cadherin into the *rhogap19d* mutant background reveals that the invading follicle cells maintain their apical adherens junctions, indicating that they have not undergone an epithelial-to-mesenchymal transition and are still epithelial in nature (Fig. 2 E).

We examined the localization of RhoGAP19D protein by using CRISPR-mediated homologous recombination to insert the mNeonGreen fluorescent tag at the N-terminus of the endogenous RhoGAP19D coding region. Neon::RhoGAP19D localizes laterally in the follicle cells, covering the full length of the domain, including the apical adherens junctions, where it sometimes appears to be slightly enriched (Fig. 2 H). A similar lateral localization was observed in all other epithelia we examined, such as the larval salivary gland, the adult testis accessory gland, the larval midgut, and the cellular blastoderm embryo (Fig. S2, A–F). Thus, RhoGAP19D seems to be a lateral factor in multiple epithelia. This is consistent with the observation that *rhogap19d* mutants die at several stages. Zygotic *rhogap19d* mutants are semilethal, with about two-thirds of homozygotes dying before adulthood. Furthermore, all embryos from homozygous mutant mothers either fail to hatch or die as first-instar larvae, indicating that it is a fully penetrant maternal effect lethal.

The *C. elegans* orthologue of RhoGAP19D, PAC-1, is recruited to cell contacts by the E-cadherin complex through redundant

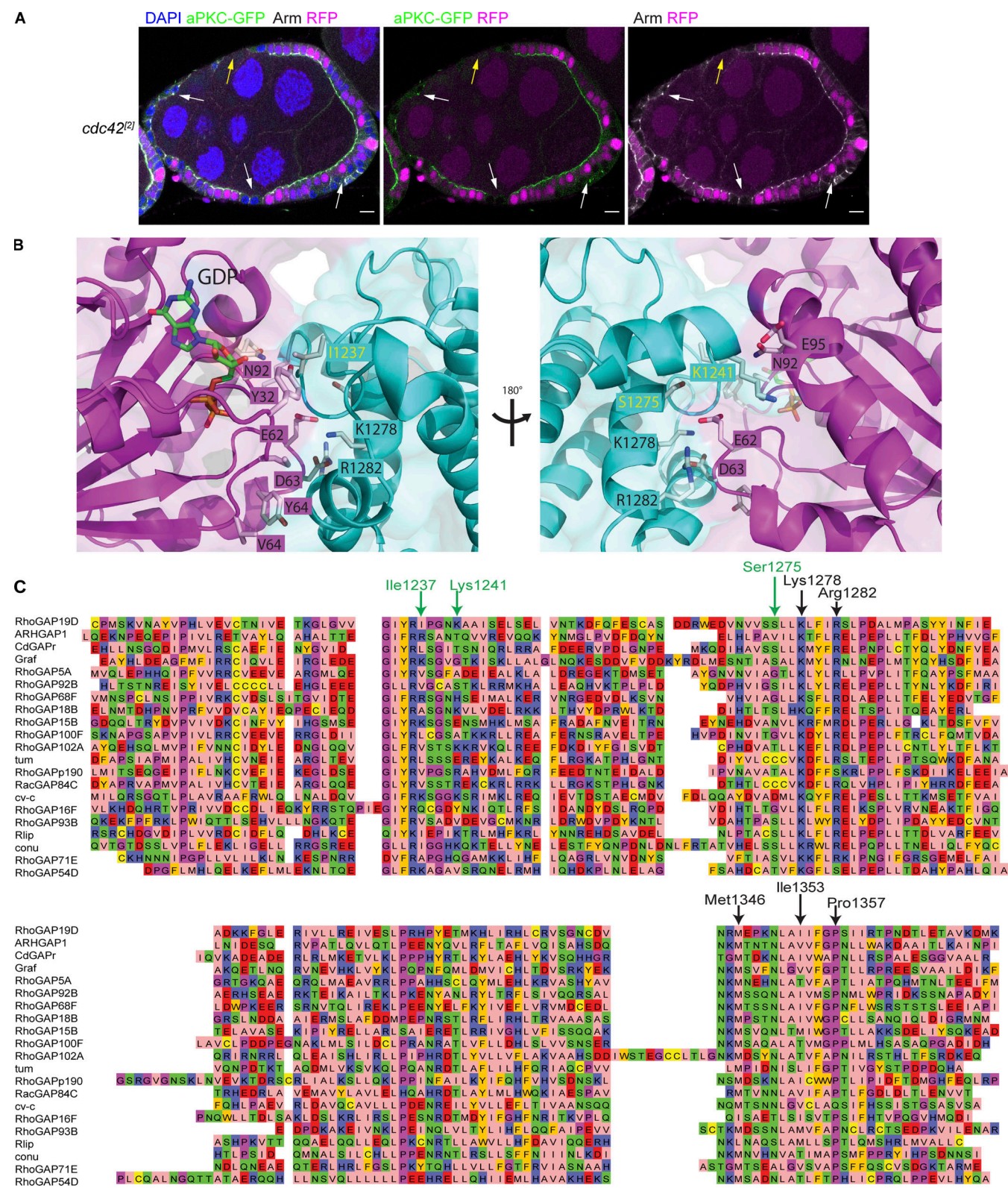

Figure 1. **Cdc42 is essential for the establishment of cell polarity in *Drosophila* follicular cells. (A)** A stage 7 egg chamber containing multiple *cdc42*[2] mutant follicular cell clones (marked by the loss of RFP; magenta) stained for Armadillo (Arm; white) and DAPI (blue) expressing endogenously tagged GFP-aPKC (green). The mutant cells are marked by arrows. GFP-aPKC is lost from the apical side of *cdc42* mutant follicular cells. In some cells, GFP-aPKC colocalizes with Armadillo in puncta. Cells lacking Cdc42 become round and often lose contact with neighboring cells, resulting in breaks in the epithelial layer. In other cases, the *cdc42* mutant cells lie basally to wild-type cells. Scale bars, 10 μm. **(B)** A diagram showing the interface between Cdc42 and bound ARHGAP1/RhoGAP19D. Key amino acids that mediate the interaction are shown in purple for Cdc42 and in green for ARHGAP1. Amino acid changes that are predicted to strengthen the interaction between Cdc42 and RhoGAP19D are shown in parentheses. **(C)** Alignment of *Drosophila* GAPs and human ARHGAP1. Conserved amino acids involved in the interaction with Cdc42 are indicated by black arrows. The green arrows mark the variable amino acids that are predicted to strengthen the interaction between Cdc42 and RhoGAP19D.

Table 1.  ***Drosophila* GTPase-activating proteins**

| Gene symbol | Gene name | Other names | UniProt accession no. |
| --- | --- | --- | --- |
| *CdGAPr* | Cd GTPase activating protein–related | GAP, d-CdGAPr | Q9VIS1 |
| *conu* | Conundrum | | Q8T0G4 |
| *cv-c* | Crossveinless c | RhoGAP88C | A8JR05 |
| *Graf* | GRAF orthologue (*Homo sapiens*) | | X2JDY8 |
| *Ocrl* | Oculocerebrorenal syndrome of Lowe | EG:86E4.5 | Q95R90 |
| *RacGAP84C* | Protein at 84C | rnRacGAP | P40809 |
| *RhoGAP1A* | Protein at 1A | EG:23E12.2 | Q6W436 |
| *RhoGAP5A* | RhoGTPase activating protein at 5A | | Q9W4A9 |
| *RhoGAP15B* | RhoGTPase activating protein at 15B | | Q0KHR5 |
| *RhoGAP16F* | RhoGTPase activating protein at 16F | | Q9VWY8 |
| *RhoGAP18B* | RhoGTPase activating protein at 18B | whir | Q9VWL7 |
| *RhoGAP19D* | RhoGTPase activating protein at 19D | | Q9VRA6 |
| *RhoGAP54D* | RhoGTPase activating protein at 54D | | A1ZAW3 |
| *RhoGAP68F* | RhoGTPase activating protein at 68F | CG 6811 | M9PC96 |
| *RhoGAP71E* | RhoGTPase activating protein at 71E | l(3)j6B9 | B7Z058 |
| *RhoGAP92B* | RhoGTPase activating protein at 92B | | A0A0B4LHC1 |
| *RhoGAP93B* | RhoGTPase activating protein at 93B | CrGAP | Q9VDE9 |
| *RhoGAP100F* | RhoGTPase activating protein at 100F | Syd-1 | Q9V9S7 |
| *RhoGAP102A* | RhoGTPase activating protein at 102A | Dm_4:1183 | H9XVN1 |
| *RhoGAPp190* | RhoGTPase activating protein at 190 | p190RhoGAP, p190 RhoGAP | Q9VX32 |
| *Rlip* | Ral interacting protein | dRalBP, D-RLIP | Q9VDG2 |
| *tum* | Tumbleweed | racGAP50C, acGAP, RacGAP, DRacGAp | Q9N9Z9 |

interactions with α-catenin and p120-catenin (Klompstra et al., 2015). We observed no change in the lateral recruitment of RhoGAP19D in *p120 catenin*–null mutants, but the junctional signal was almost completely lost when α-catenin was depleted by RNAi (Fig. 2, I–L). Thus, RhoGAP19D is localized to the lateral membrane by a nonredundant interaction with α-catenin, which links it to cadherin adhesion complexes. The lateral localization of RhoGAP19D was strongly reduced in clones homozygous for *shotgun*[IG29], a null mutant in *shotgun* (E-cadherin), whereas clones homozygous for a deletion of N-cadherin 1 and N-cadherin 2 had no effect (Figs. 2 M and S2 K; Tepass et al., 1996; Prakash et al., 2005). The weaker phenotype of *shotgun*[IG29] clones compared with α-catenin knockdown is presumably because N-cadherin is upregulated in E-cadherin mutants, and either E- or N-cadherin can recruit α-catenin and RhoGAP19D (Grammont, 2007). α-Catenin and E-cadherin are concentrated in the apical adherens junctions, whereas RhoGAP19D shows only a slight apical enrichment and is much more uniformly distributed along the lateral membrane. Because this suggests that other factors may modulate the recruitment of RhoGAP19, we examined whether any lateral polarity factors affect its distribution, but we observed no change when the lateral adhesion proteins, FasII, FasIII, or Neuroglian, were knocked down by RNAi or in null mutant clones for the lateral polarity factors Lgl, Scribble (Scrib), and Coracle (Fig. 2 N; and Fig. S2, H–N).

The localization of RhoGAP19D suggests that it may function to inhibit Cdc42 laterally. We therefore examined where Cdc42 is active by following the localization of an endogenously tagged version of the Cdc42 effector, N-WASP (Kim et al., 2000). N-Wasp-Neon is expressed at very low levels in the follicle cells, with slightly higher expression in the posterior cells. In wild-type cells, N-WASP-Neon localizes exclusively to the apical domain, consistent with the apical localization of active Cdc42. By contrast, N-WASP also localizes along the lateral membrane in *rhogap19d* mutant clones at the posterior (Fig. 3 A). Although N-Wasp-Neon is harder to detect in lateral follicle cells, horizontal sections through regions containing clones also reveal lateral localization in the mutant cells, but not in wild-type cells (Fig. 3 B). Thus, RhoGAP19D is required to exclude active Cdc42 from the lateral domain.

To confirm that RhoGAP19D represses Cdc42 activity, we used upstream activating sequence (UAS)-GrabFP-A$_{int}$ to mislocalize the protein to the apical domain (Harmansa et al., 2017). The GrabFP-A$_{int}$ construct consists of an N-terminal Cherry, a transmembrane domain, and an anti-GFP nanobody fused to the localization signal of Baz (Par-3; Fig. 3 C). When this construct is expressed in the follicle cells under the control of Tj-Gal4, the fusion protein localizes to the apical membrane and apical junctions without any apparent effect on the appearance of the cells (Fig. 3 D). Similarly, overexpression of UAS-GFP-RhoGAP19D alone results in higher levels of RhoGAP19D along the

**Table 2.** *Drosophila* GTPase-activating proteins ranked by the predicted strength of their interactions with Cdc42

| Protein 1 | Gene 1 | % Id 1 | Protein 2 | Gene 2 | % Id 2 | PDB accession no. | Z-score |
|---|---|---|---|---|---|---|---|
| Q9VRA6-RhoGAP | RhoGAP19D | 24 | P40793-RAS | Cdc42 | 94 | 1AM4 | 3.329 |
| M9PC96-RhoGAP | RhoGAP68F | 40 | P40793-RAS | Cdc42 | 92 | 1GRN | 3.122 |
| Q9VIS1-RhoGAP | CdGAPr | 26 | P40793-RAS | Cdc42 | 92 | 1GRN | 2.999 |
| A0A0B4LHC1-RhoGAP | RhoGAP92B | 36 | P40793-RAS | Cdc42 | 92 | 2NGR | 2.9 |
| P40809-RhoGAP | RacGAP84C | 26 | P40793-RAS | Cdc42 | 92 | 1GRN | 2.88 |
| Q8T0G4-RhoGAP | conu | 26 | P40793-RAS | Cdc42 | 92 | 1GRN | 2.835 |
| Q9VDE9-RhoGAP | RhoGAP93B | 30 | P40793-RAS | Cdc42 | 92 | 1GRN | 2.823 |
| X2JDY8-RhoGAP | Graf | 29 | P40793-RAS | Cdc42 | 94 | 1AM4 | 2.748 |
| Q0KHR5-RhoGAP | RhoGAP15B | 28 | P40793-RAS | Cdc42 | 94 | 1AM4 | 2.723 |
| A8JR05-RhoGAP | cv-c | 29 | P40793-RAS | Cdc42 | 94 | 1AM4 | 2.642 |
| Q9VWL7-RhoGAP | RhoGAP18B | 28 | P40793-RAS | Cdc42 | 94 | 1AM4 | 2.488 |
| Q9VWY8-RhoGAP | RhoGAP16F | 22 | P40793-RAS | Cdc42 | 94 | 1AM4 | 2.142 |
| Q9N9Z9-RhoGAP | tum | 22 | P40793-RAS | Cdc42 | 94 | 1AM4 | 1.951 |
| Q9VX32-RhoGAP | RhoGAPp190 | 28 | P40793-RAS | Cdc42 | 92 | 1GRN | 1.93 |
| Q9VDG2-RhoGAP | Rlip | 32 | P40793-RAS | Cdc42 | 92 | 2NGR | 1.683 |

lateral membrane but has no effect on cell polarity or morphology during stages 1–8 of oogensis. When GFP-RhoGAP19D and GrabFP-A$_{int}$ are coexpressed, however, the apical recruitment of GFP-RhoGAP19D by the anti-GFP nanobody disrupts polarity and epithelial organization, as shown by the failure to concentrate aPKC apically and the irregular cell shapes (Fig. 3 E). This phenotype closely resembles that of *cdc42* mutants, providing further evidence that RhoGAP19D is a specific Cdc42GAP.

To investigate the cellular basis for the invasive behavior of *rhogap19d* mutant follicular cells, we compared the phenotypes of mutant and wild-type cells in the same epithelium by generating homozygous mutant clones. Live imaging revealed that *rhogap19d* mutant cells are taller than wild-type cells, with dome-shaped apical surfaces that protrude into the germline (Fig. 4 A). The distance between the apical and basal surfaces of mutant cells is 20% greater than in wild type in fixed samples (Fig. 4, B and F). This is compensated for by a 15% reduction in cell width, indicating that cell volume is, if anything, slightly reduced (Fig. 4 G). The adherens junctions, marked by E-cadherin–GFP and Canoe (Cno), do not show a corresponding change in position and remain level with or slightly below the adherens junctions in the neighboring wild-type cells (Fig. 4, B and C). The adherens junctions mark the boundary between the apical and lateral domains, suggesting that the apical domain has expanded. This is indeed the case, because GFP-aPKC localizes all over the domed region of membrane above the adherens junctions, which is 40% longer than in wild type (Fig. 4, C and H). The apical transmembrane protein Crumbs shows a similar extension across the expanded apical domain but is more enriched in the subapical region above the adherens junctions, reflecting its accumulation in regions where it can engage in homophilic interactions with Crumbs in adjacent cells (Fig. 4 D; Thompson et al., 2013). By contrast, the lateral domain, marked by Lgl-GFP, is decreased in length (Fig. 4, E and I). Lateral Cdc42

activity therefore expands the apical domain at the expense of the lateral domain to generate taller cells that protrude into the germline. A similar apical expansion is also observed in *rhogap19d* mutant testis accessory glands and in the primary epithelium of cellular blastoderm embryos derived from *rhogap19d* mutant mothers, suggesting that this is a general phenotype of loss of RhoGAP19D in *Drosophila* epithelia (Fig. S3).

To gain insight into how *rhogap19d* mutant follicle cells invade the germline, we imaged living egg chambers at stages 5–7, the stages when invasions are most likely to occur (Video 1; Fig. 5 A). The mutant cells are not only taller than wild-type cells with domed apical surfaces but are also more motile. Temporal projections show that the mutant cells expand and contract along their apical–basal axes, whereas wild-type cells are static (Fig. 5 B). The apical expansion of the mutant cells and the up and down movements are likely to increase strain in the epithelium and raise the probability of regions of the follicle cell layer invading the germline (Fig. 5 A). More rarely, we observed clusters of cells that had detached from the basement membrane and were beginning to invade (Fig. 5 C).

The higher motility suggests that myosin activity is increased in mutant cells, and we therefore examined the distribution of nonmuscle myosin II (NMYII) using a protein trap insertion in the heavy chain (Zipper). This revealed that the mutant cells have more numerous and larger NMYII foci along their lateral membranes and reduced levels of apical NMYII (Fig. 5 D). This increase in lateral NMYII is likely to account for the apical–basal contractions in mutant cells. In MDCK cells, Cdc42 recruits and activates NMYII apically through its effector, MRCK, which phosphorylates the myosin regulatory light chain to stimulate contractility (Zihni et al., 2017; Zhao and Manser, 2015). This suggests that the *Drosophila* orthologue of MRCK, Gek, might play a similar role in coupling Cdc42 to the activation of NMYII in the follicular epithelium. Antibody staining revealed that Gek

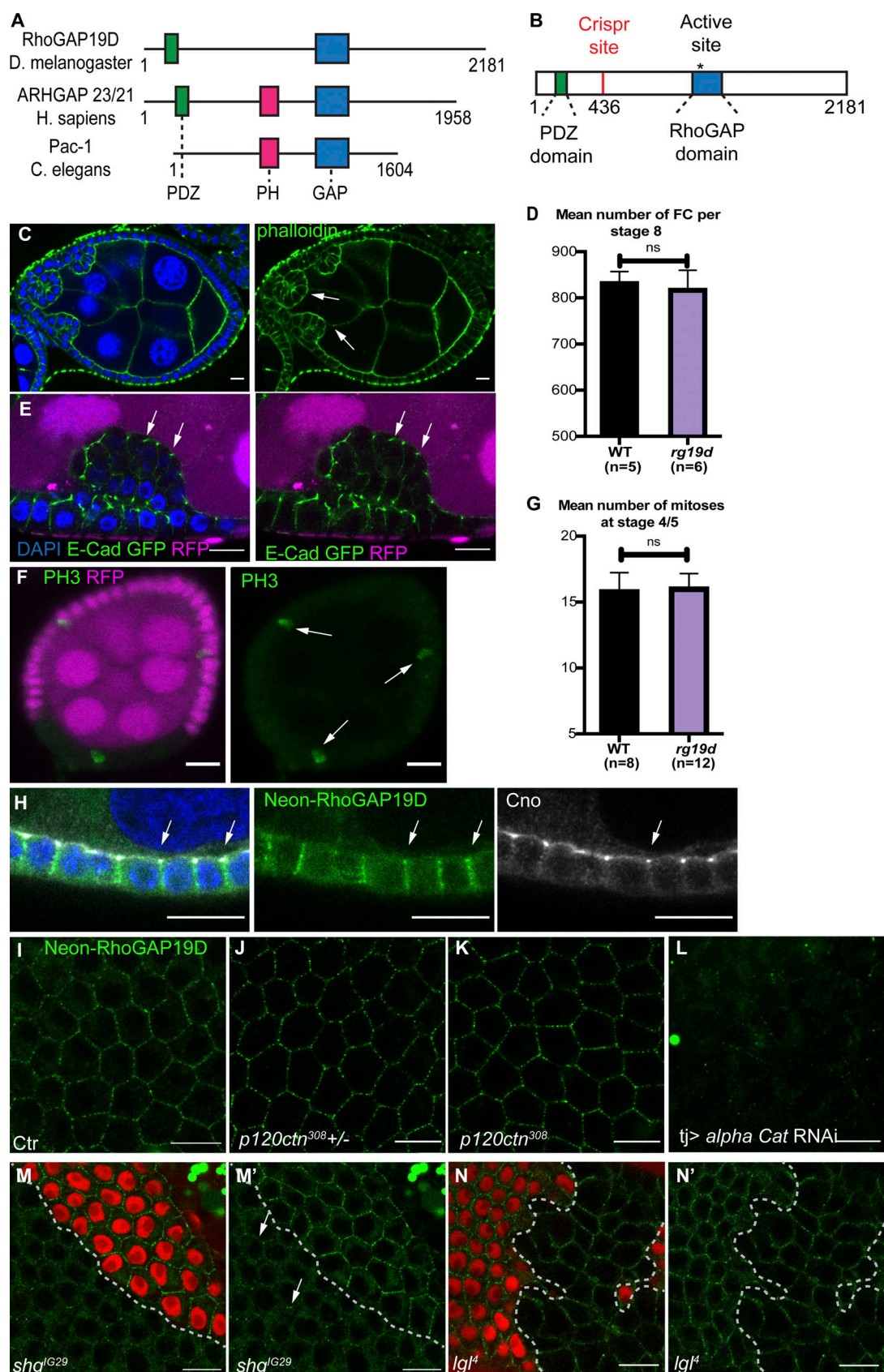

Figure 2. **RhoGAP19D is required for the integrity of the epithelial layer. (A)** Comparison of the domain structure of *Drosophila* RhoGAP19D with its orthologues, human ARHGAP23/21 and *C. elegans* PAC-1. RhoGAP19D contains PDZ and GAP domains but lacks a PH domain. **(B)** Diagram showing the CRISPR-induced mutations in RhoGAP19D. The mutations generate proteins that lack the GAP domain but still contain the PDZ domain. **(C)** A stage 7 *rhogp19d* mutant

egg chamber showing invasion of the follicular epithelium into the adjacent germline. The mutant cells (white arrows) invade between the nurse cells at the anterior of the egg chamber. Phalloidin staining is shown in green and DAPI in blue. **(D)** Graph showing the number of wild-type and *rhogap19d* mutant follicle cells (FC) per egg chamber at stage 8. **(E)** Invading *rhogap19d* mutant cells (marked by the loss of RFP; magenta) maintain their adherens junctions (white arrows) and epithelial organization, as shown by E-cadherin–GFP expression (DAPI; blue). **(F)** PH3 staining (green) of mitotic cells (white arrows) in a stage 5 egg chamber containing a *rhogap19d* mutant clone (marked by the loss of RFP; magenta). There is no increase in cell division in the mutant clone. **(G)** Quantification of the number of mitoses in wild-type and *rhogap19d* mutant egg chambers during stages 4 and 5. **(H)** mNeonGreen-RhoGAP19D (green) localizes to the lateral domain of the follicle cells and is slightly enriched at the adherens junctions (white arrows) stained with Cno (white) and DAPI (blue). RhoGAP19D protein in the germline was depleted by RNAi. **(I)** Surface view of wild-type cells expressing mNeonGreen-RhoGAP19D. **(J and K)** mNeonGreen-RhoGAP19D localizes normally in *p120 catenin*[308]/+ and *p120 catenin*[308] homozygous cells. **(L and M)** mNeonGreen-RhoGAP19D recruitment to lateral domain is almost lost in cells expressing α-catenin RNAi (L) and is strongly reduced (white arrows) in *shotgun*[IG29] mutant clones (marked by the loss of RFP; magenta; M and M′). **(N and N′)** mNeonGreen-RhoGAP19D localization is not affected in *lgl*[4] mutant clones (marked by the loss of RFP; magenta) compared with wild-type cells. Scale bars, 10 µm.

is predominantly localized to the apical surface of the follicle cells, consistent with its role in MDCK cells (Fig. 5 E). Gek extends along the lateral membrane, however, in all *rhogap19d* mutant cells (Fig. 5 F). Thus, the ectopic Cdc42 activity in *rhogap19d* mutants recruits Gek to the lateral cortex, where it can localize and activate NMYII.

Our results suggest that the invasive phenotype of *rhogap19d* mutants depends on a partial disruption of polarity, in which the apical domain expands at the expense of the lateral domain. Because the relative sizes of the apical and lateral domains are determined by mutual antagonism between apical and lateral polarity factors, reducing the dosage of lateral factors should enhance this phenotype, whereas reducing apical factors should suppress it. We therefore tested whether polarity mutants act as dominant modifiers of the *rhogap19d* phenotype (Fig. 6). Removing one copy of the lateral polarity proteins, *lgl* and *scrib*, doubles the frequency of germline invasion, as does removing both copies of *fasciclin II* or RNAi-knockdown of *neuroglian*, both of which encode lateral adhesion factors (Bilder and Perrimon, 2000; Wei et al., 2004; Szafranski and Goode, 2007). By contrast, loss of one copy of *aPKC* or *crb* strongly suppresses invasion. Reducing the dosage of *gek* also decreases the frequency of invasion, consistent with its role in activating NMYII laterally to stimulate the movement of the follicular cells into the germline. Thus, these genetic interactions support the view that the invasive behavior of *rhogap19d* mutant cells is driven by the expansion of the apical domain and Gek-dependent lateral contractility, both of which will increase the stress on the epithelium without completely disrupting polarity.

Two of the mutants showed unexpected genetic interactions with *rhogap19d*. First, reducing the dosage of the lateral polarity factor, Par-1, suppressed the invasive phenotype of the *rhogap19d* mutant, whereas the other lateral factors strongly enhance it. Par-1 localizes to the lateral membrane and functions to limit the basal extent of the adherens junctions by phosphorylating and antagonizing Baz (Par-3; Benton and St Johnston, 2003; Wang et al., 2012). The ability of the *par-1* mutant to suppress *rhogap19d* indicates that Par-1 does not function in the same pathway as Scrib, Lgl, FasII, and Nrg and suggests instead that it either negatively regulates these lateral factors or positively regulates apical ones. It is also possible that Par-1 acts through the actin cytoskeleton and is required for the lateral contractility induced by ectopic Gek activity. Second, p21-activated kinase 1 (Pak1) has been reported to function redundantly with aPKC to specify the

apical domain downstream of active Cdc42 (Aguilar-Aragon et al., 2018). Although one would therefore expect the *pak1* mutant to suppress the invasive phenotype–like mutants in the other apical factors, it acts as a strong enhancer of invasion. This is consistent with the role of Pak1 as a component of the lateral Scribble complex and argues against the proposal that it functions as an apical Cdc42 effector kinase (Bahri et al., 2010).

## Discussion

Here we report that RhoGAP19D restricts Cdc42 activity to the apical side of the follicle cells and probably many other *Drosophila* epithelial tissues. In the absence of RhoGAP19D, both N-WASP and Gek are recruited to the lateral membrane, indicating that Cdc42 is ectopically activated there. This implies that RhoGAP19D is the major Cdc42GAP that represses Cdc42 laterally, because no other GAPs can compensate for its loss. This also suggests that the GEFs that activate Cdc42 are not restricted to the apical domain and can turn it on laterally once this repression is removed. This is consistent with the identification of multiple vertebrate GEFs with different localizations that contribute to apical Cdc42 activation (Otani et al., 2006; Qin et al., 2010; Rodriguez-Fraticelli et al., 2010; Oda et al., 2014; Zihni et al., 2014). Our results therefore identify RhoGAP19D as a new lateral polarity factor. This leads to a revised network of polarity protein interactions in which RhoGAP19D functions as the third lateral factor that antagonizes the activity of apical factors, alongside Lgl, which inhibits aPKC, and Par-1, which excludes Baz/Par-3 (Fig. 7 A; Wirtz-Peitz et al., 2008; Benton and St Johnston, 2003).

The function of RhoGAP19D is very similar to that of its orthologue PAC-1, which inhibits Cdc42 at sites of cell contact in early *C. elegans* blastomeres to generate distinct apical and basolateral domains (Anderson et al., 2008). Both RhoGAP19D and PAC-1 are recruited to the lateral domain by E-cadherin complexes, although the exact mechanism is slightly different. RhoGAP19D recruitment is strictly dependent on α-catenin, which links it through β-catenin to the E-cadherin cytoplasmic tail, whereas α-catenin (HMP-1) and p120-catenin (JAC-1) play partially redundant roles in recruiting PAC-1 to E-cadherin (HMR-1) in the worm (Klompstra et al., 2015). Nevertheless, in both cases, the recruitment of the Cdc42GAP translates the spatial cue provided by the localization of cadherin to sites of cell–cell contact into a polarity signal that distinguishes the

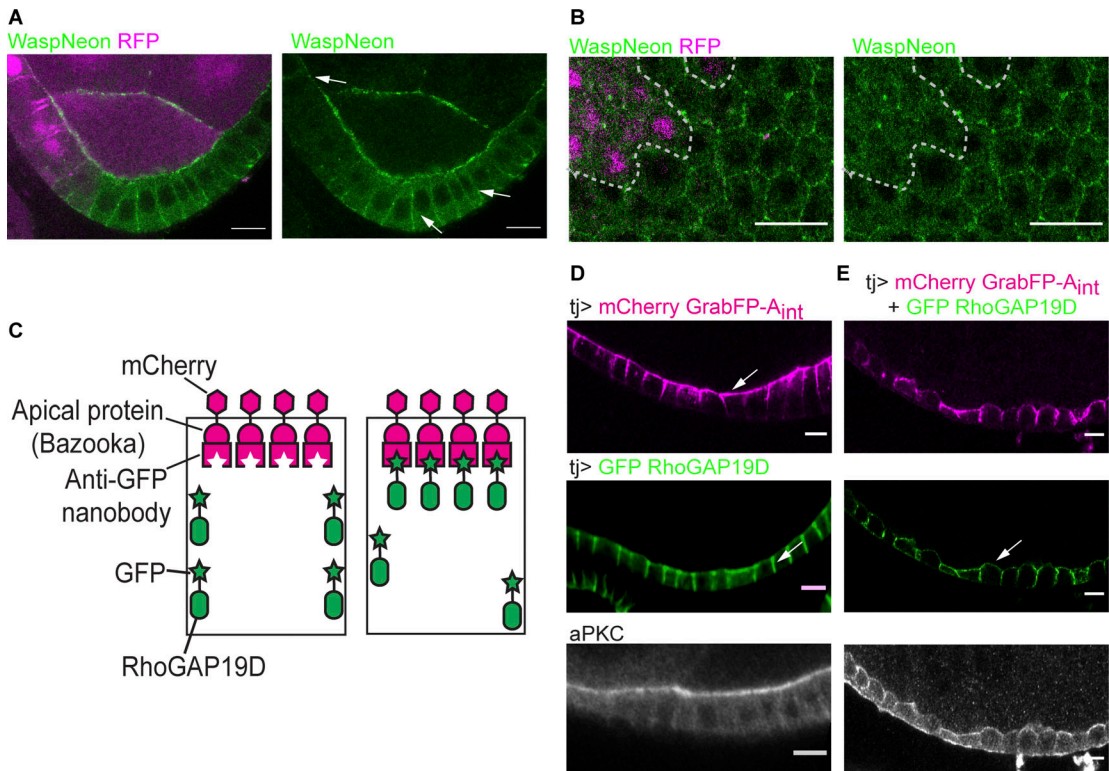

**Figure 3. RhoGAP19D inhibits CDC42 activity laterally. (A)** The CDC42 effector, N-WASP (tagged with mNeonGreen), spreads laterally in *rhogap19D* mutant cells (marked by the loss of RFP; magenta, white arrows). This phenotype was observed in 20 of 27 mutant cells. **(B)** A horizontal section through a lateral region of the follicular epithelium containing a *rhogap19D* mutant clone marked by the loss of RFP (red), expressing mNeonGreen- N-WASP. N-WASP localizes to the lateral membrane of the mutant cells. **(C)** Cartoon showing the UAS-GrabFP-A$_{int}$ system. **(D)** mCherry-GrabFP-A$_{int}$ predominantly localizes to the apical side of the follicular cells when expressed under the control of TJ-Gal4, whereas GFP-RhoGAP19D alone localizes laterally (white arrow). **(E)** Coexpression of mCherry-GrabFP-A$_{int}$ and GFP-RhoGAP19D results in the apical recruitment of RhoGAP19D, leading to a loss of epithelial polarity and mislocalization of aPKC (white arrow). Scale bars, 10 μm.

lateral from the apical domain. Classic work on the establishment of polarity MDCK cells grown in suspension has revealed that the recruitment of cadherin (uvomorulin) to sites of cell–cell contact is the primary cue that drives the segregation of apical proteins from basolateral proteins (Wang et al., 1990). Furthermore, the expression of E-cadherin in unpolarized mesenchymal cells is sufficient to induce this segregation, although the mechanisms behind this process are only partially understood (Wang et al., 1990; McNeill et al., 1990; Watabe et al., 1994; Nejsum and Nelson, 2007). Our observation that RhoGAP19D directly links cadherin adhesion to the polarity system in epithelial cells extends the results of Klompstra et al. (2015) in early blastomeres, strongly suggesting that PAC-1/RhoGAP19D plays an important role in the first steps in epithelial polarization.

Although PAC-1 and RhoGAP19D perform equivalent functions in early blastomeres and epithelial cells, there is one important difference between their mutant phenotypes. In *pac-1* mutants, Par-6 and aPKC are mislocalized to the contacting surfaces of *C. elegans* blastomeres where Cdc42 is ectopically active (Anderson et al., 2008). By contrast, Par-6 and aPKC are not mislocalized laterally in *rhogap19d* mutant *Drosophila* epithelial cells, even though lateral Cdc42-GTP does recruit two other Cdc42 effectors, N-WASP and Gek. Thus, lateral Cdc42 activity is sufficient to recruit Par-6/aPKC to the lateral domain

in early blastomeres, but not in epithelial cells. Instead, we observed that lateral Cdc42 activity in *rhogap19d* mutant follicle cells acts at a distance to expand the size of the apical domain. A likely explanation for this difference is the presence of Crumbs in epithelial cells. The interaction between Cdc42-GTP and Par-6 alters the conformation of Par-6 so that it can bind to Crumbs, which anchors the Par-6–aPKC complex to the apical membrane and activates aPKC's kinase activity (Peterson et al., 2004; Whitney et al., 2016; Dong et al., 2020). Although Par-6 presumably binds to Cdc42 laterally in *rhogap19D* mutants and undergoes the conformational change, it cannot be anchored laterally in the absence of Crumbs. This activated Par-6–aPKC complex can then diffuse until it is captured by Crumbs in the apical domain, thereby increasing apical aPKC activity, providing an explanation for why the apical domain expands in *rhogap19d* mutant cells (Fig. 7 B). *C. elegans* has three Crumbs orthologues, but removal of all three simultaneously has no effect on viability or polarity (Waaijers et al., 2015). Thus, in contrast to *Drosophila* epithelial cells, *C. elegans* Crumbs proteins are not required for Par-6/aPKC localization and activation, suggesting that some other mechanism, such as Cdc42 binding, is sufficient to activate aPKC.

If the failure of active Cdc42 to recruit aPKC laterally in *rhogap19d* mutant cells is due to the absence of Crumbs in this

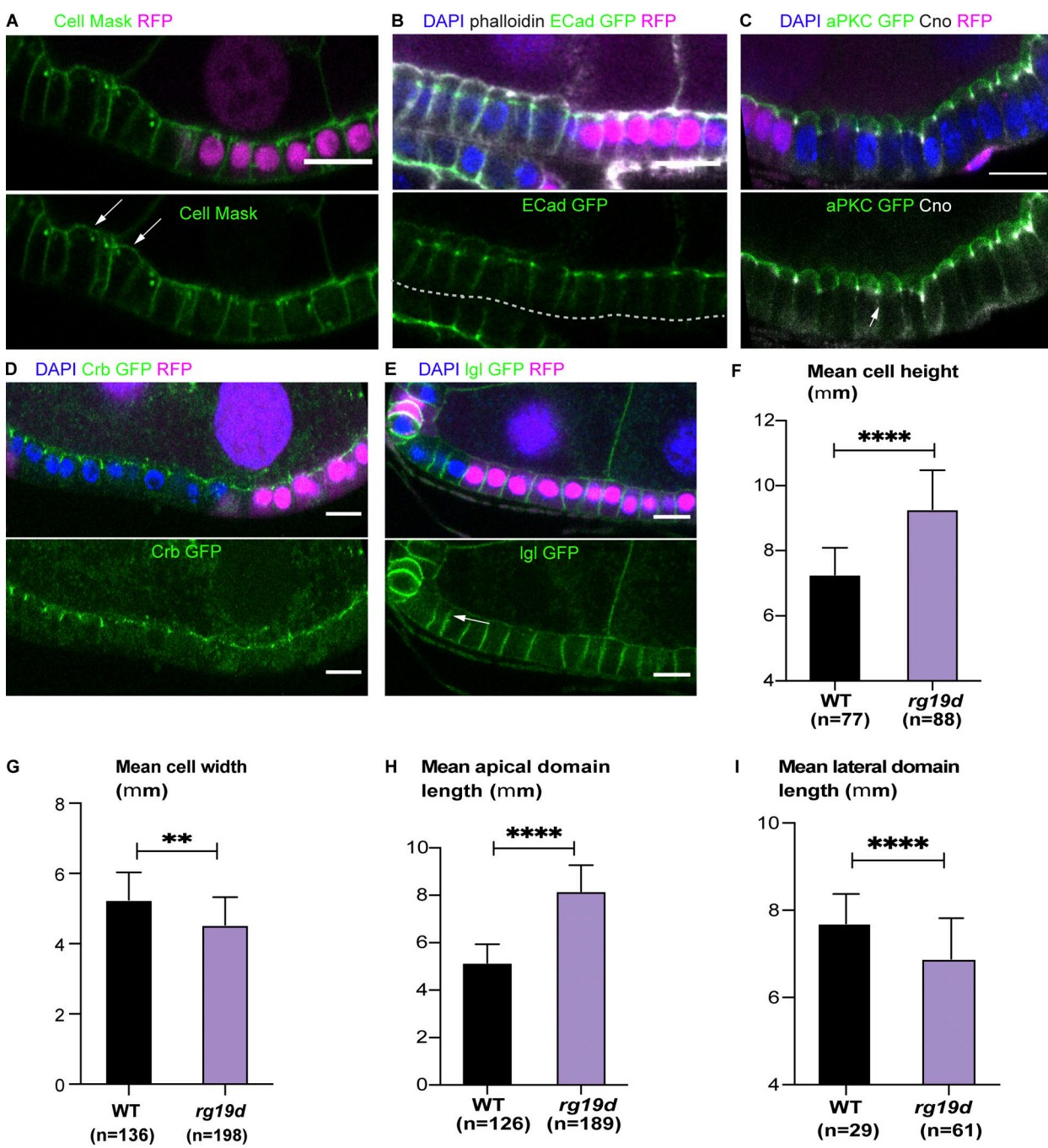

Figure 4. ***rhogap19D* mutant cells are taller than wild-type cells and have an enlarged apical domain. (A–E)** Regions of the stage 7 follicle cell epithelium stained with DAPI (blue) containing clones of *rhogap19d* mutant cells marked by the loss of RFP (magenta). **(A)** Cell mask staining (green) of the plasma membranes reveals that mutant cells are taller than wild-type cells and have domed apical surfaces. The arrows indicate the apical surfaces of the mutant cells. **(B)** The adherens junctions marked by endogenously tagged E-cadherin (ECad)-GFP (green) form at the same level in *rhogap19d* mutant and wild-type cells (phalloidin; white). **(C)** In *rhogap19d* mutant cells, GFP-aPKC localizes all around the apical domain above the adherens junctions (marked by Cno staining; white; indicated by the white arrow). **(D)** Crb-GFP marks an enlarged subapical region in *rhogap19d* cells. **(E)** *rhogap19d* mutant cells have slightly shorter lateral domains than wild-type cells, as shown by Lgl-GFP localization (green; white arrow). **(F)** A graph showing the mean cell height in wild-type and *rhogap19d* mutant cells. **(G)** A graph showing the mean cell width in wild-type and *rhogap19d* mutant cells. **(H)** A graph showing the mean apical domain length in wild-type and *rhogap19d* mutant cells. **(I)** A graph showing the mean lateral domain length in wild-type and *rhogap19d* mutant cells. The error bars represent SEM; ****, $P < 0.0001$; **, $P < 0.002$. Scale bars, 10 µm.

region, there must be a mechanism to exclude Crumbs from the lateral domain. One proposed mechanism depends on Yurt (Moe and EPB41L5 in vertebrates), which is restricted to the lateral domain by aPKC and binds to Crumbs to antagonize its activity (Laprise et al., 2006). However, we did not observe any lateral recruitment of aPKC in *rhogap19d;yurt* double-mutant cells. Thus, there must be some parallel mechanism that excludes Crumbs, Par-6, and/or aPKC from the lateral domain.

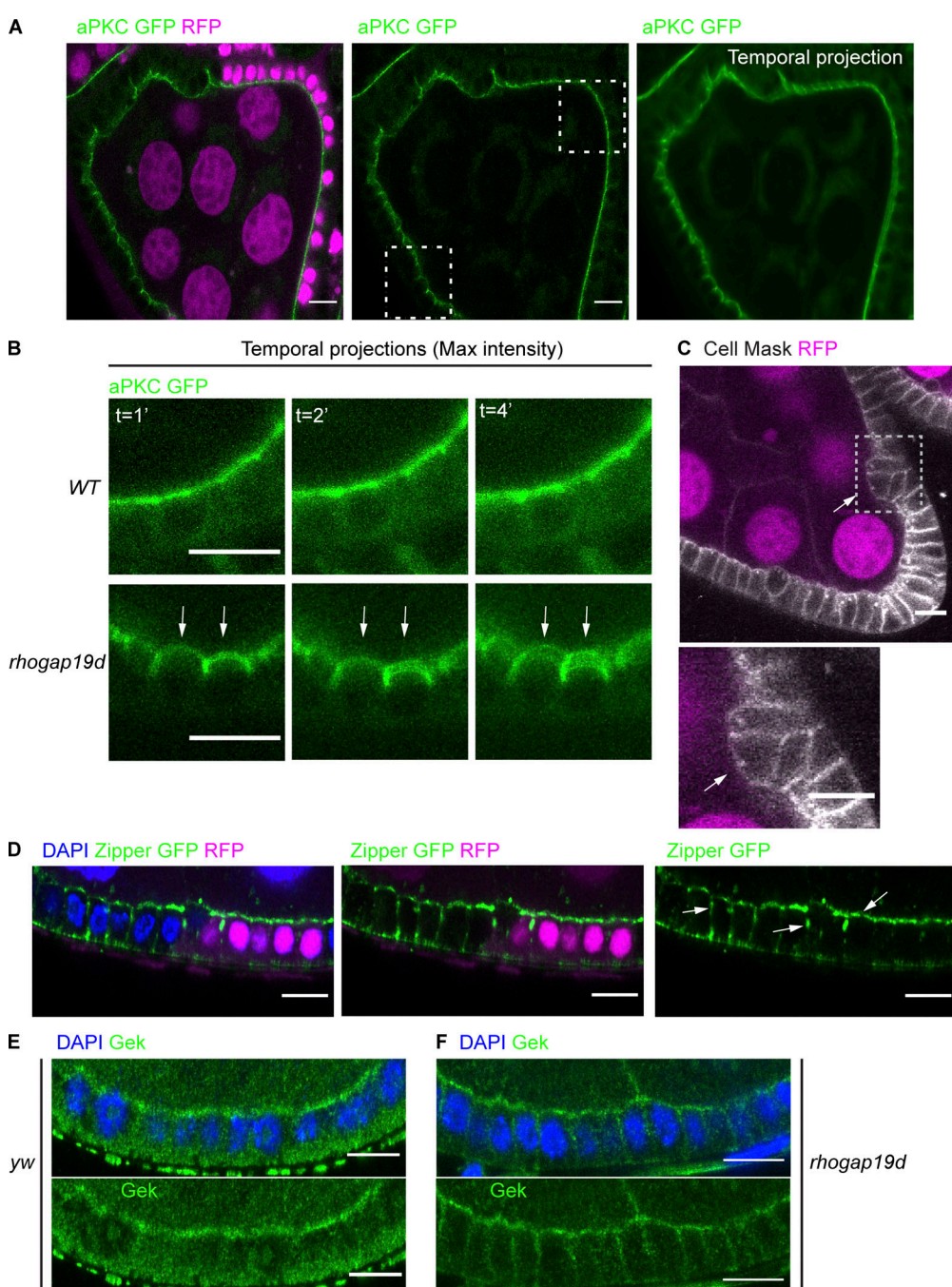

**Figure 5.** ***rhogap19d* mutant cells are more motile than wild-type cells and contract laterally. (A)** Single frames and a temporal projection of a time-lapse movie of a stage 7 egg chamber containing a large *rhogap19d* mutant clone (marked by the loss of RFP; magenta) and expressing GFP-aPKC (green). The blurred apical surfaces of the mutant cells in the temporal projection indicate that they are moving between frames. **(B)** Magnification of the boxed areas in A, showing that *rhogap19d* mutant cells (bottom panels) are more motile (white arrows) than wild-type cells (top panels). Temporal projections after 1, 2, and 4 min. **(C)** A single frame from a movie showing a cluster of *rhogap19d* mutant cells (white arrow; marked by the loss of RFP; magenta) beginning to invade the germline. The cells in the cluster appear to have detached from the basement membrane (cell mask; white). The lower panel shows a magnification of the boxed area. **(D)** *rhogap19d* mutant cells have lateral foci of NMYII foci (Zipper-GFP; green) and reduced levels at the apical side compared with wild-type cells (white arrows; DAPI, blue). This phenotype was observed in 154 of 157 mutant cells. **(E and F)** Gek (green) localizes apically in wild-type follicle cells (E) but extends along the lateral domain of all *rhogap19d* mutant cells (F; DAPI, blue). Scale bars, 10 μm. *n* = 11 homozygous mutant egg chambers.

Although loss of RhoGAP19D only leads to a partial disruption of polarity, it causes the follicular epithelium to invade the adjacent germline tissue with 40% penetrance. This invasive behavior is not driven by an epithelial-to-mesenchymal transition, because the cells retain their apical adherens junctions and epithelial organization. Instead, the deformation of the epithelium seems to be driven by the combination of an increase in lateral contractility and an expansion of the apical domain, because

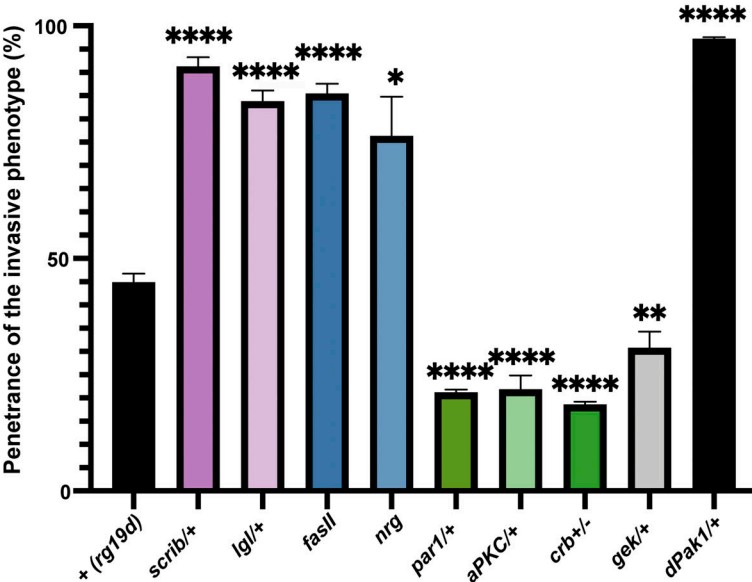

**Genetic interactions between *rhogap19d* and other apical/ lateral factors**

| | **(rg19d)** | **scrib/+** | **lgl/+** | **fasII** | **nrg** | **par1/+** | **aPKC/+** | **crb/+** | **gek/+** | **dPak1/+** |
|---|---|---|---|---|---|---|---|---|---|---|
| egg chambers analyzed | 301 | 194 | 94 | 185 | 114 | 155 | 98 | 129 | 78 | 228 |
| egg chambers with invasions | 132 | 176 | 78 | 158 | 86 | 32 | 21 | 23 | 23 | 221 |
| penetrance % | 44 | 91 | 83 | 85 | 76 | 21 | 22 | 18 | 30 | 97 |
| P-value (significantly different if P<0.05) | | <0.0001 | <0.0001 | <0.0001 | 0.02 | <0.0001 | <0.0001 | <0.0001 | 0.001 | <0.0001 |

Figure 6. **Genetic interactions between *rhogap19d* and other polarity factors.** A histogram showing the penetrance of the germline invasion phenotype of large *rhogap19d* mutant clones in combination with other polarity mutants. Removing one copy of *scrib*, *lgl*, or *Pak1* strongly enhances the penetrance of the invasion phenotype. *rhogap19d;fasII* double-mutant clones and *rhogap19d* clones in which *nrg* has been depleted by RNAi also show a highly penetrant invasive phenotype. Loss of one copy of *aPKC*, *gek*, *crb*, or *par-1* strongly reduces the frequency of invasions. The error bars represent SEM; *, P < 0.05; **, P < 0.002; ****, P < 0.0001. All numerical data are presented in the table.

reducing the dosage of Gek, which activates myosin II to drive the contractility, significantly reduces the frequency of this phenotype, as does halving the dosage of any of the apical polarity factors. The expansion of the apical domain makes the domain too long for the cells to adopt the lowest-energy conformation, giving them a tendency to become wedge shaped, which could drive the evagination. It is also possible that buckling of the epithelium contributes to invasion. Recent work has shown that epithelial monolayers under compressive stress and constrained by a rigid external scaffold have a tendency to buckle inward (Trushko et al., 2020). The follicular cell layer is surrounded by an ECM that constrains the shape of the egg chamber and that should therefore resist expansion (Haigo and Bilder, 2011). In addition, the pulses of lateral contractility are likely to generate compressive stress because transiently reducing cell height while maintaining a constant volume will increase the cells' cross-sectional area, thereby exerting a pushing force on the neighboring cells. This compression coupled to the tendency to become wedge shaped due to apical expansion could therefore trigger the rare buckling events that initiate invasion. In support of this view, lateral contractility has been shown to drive the folding of the imaginal wing disc between the prospective hinge region and the

pouch (Sui et al., 2018). This phenotype provides an example of how a partial disruption of polarity can induce cell shape changes that lead to major alterations in tissue morphogenesis (St Johnston and Sanson, 2011).

The *rhogap19d* phenotype resembles the defects earliest observed in the development of ductal carcinoma in situ (Halaoui et al., 2017). In flat epithelial atypia (FEA), the ductal cells are still organized into an epithelial layer, but they display apical protrusions that are strongly labeled by the apical polarity factor Par-6. This suggests that the apical domain has expanded and bulges out of the cell, just as we observed in the *rhogap19d* mutant follicular cells. In the next stage, atypical ductal hyperplasia (ADH), the ductal cells start to invade the lumen of the duct while retaining aspects of normal apical–basal polarity (Fig. S4 A). This again resembles the invasive phenotype of *rhogap19d* mutants, although overproliferation of the ductal cells probably also contributes to invasion in this case. Thus, these abnormalities, which can sometimes progress to ductal carcinoma in situ and breast cancer, mirror the effects of lateral Cdc42 activation. The RhoGAP19D human orthologues, ARHGAP21 and ARHGAP23, have been shown to bind directly to α-catenin and localize to cell–cell junctions (Sousa et al., 2005; Van Itallie et al., 2014).

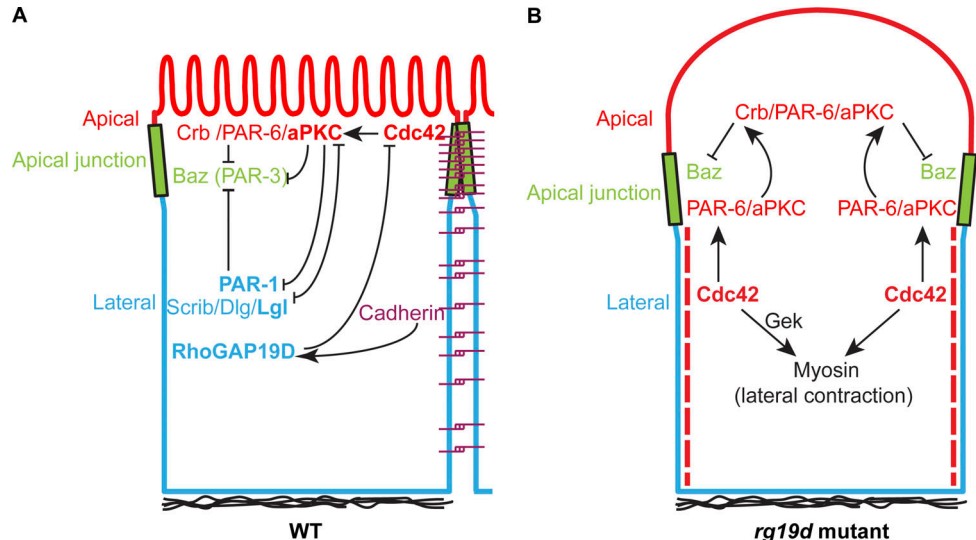

Figure 7. **A revised network of inhibitory interactions between polarity factors. (A)** A diagram showing how the recruitment of RhoGAP19D to the lateral membrane by E-cadherin adhesion complexes restricts Cdc42 activity to the apical domain. This adds a new inhibitory interaction to the network of interactions between apical and lateral polarity factors. **(B)** A model of the changes in *rhogap19d* mutant cells that lead to the invasive phenotype. In the absence of RhoGAP19D, active Cdc42 localizes along the lateral domain as well as the apical domain and activates Gek to induce lateral actomyosin contractions. Lateral Cdc42-GTP also alters the conformation of Par-6/aPKC so that it is competent to bind to Crumbs. This primed Par-6/aPKC then diffuses until it binds to apical Crumbs, which activates aPKC's kinase activity, resulting in expansion of the apical domain and a dome-shaped apical surface.

Furthermore, low expression of ARHGAP21 or ARHGAP23 correlates with reduced survival rates in several cancers of epithelial origin (Fig. S4, C and D; Györffy et al., 2010). It would therefore be interesting to determine whether these orthologues perform the same functions in epithelial polarity as RhoGAP19D and if their loss contributes to tumor development.

## Materials and methods

### Predicting Cdc42–GAP interactions
We identified and aligned putative *Drosophila* GAPs by searching the Pfam (Finn et al., 2016) hidden Markov model profile PF00620.26 (RhoGAP) against the UniProt *Drosophila melanogaster* proteome using HMMsearch (Eddy, 2009). We aligned significantly scoring sequences together with human ARHGAP1 (from the 3D structure Research Collaboratory for Structural Bioinformatics [RCSB] Protein Data Bank accession no. 1GRN) using HMMalign (Eddy, 2009).

We scored the potential interaction of each *Drosophila* GAP/Cdc42 pair using the structure of human ARHGAP1/CDC42 (RCSB Protein Data Bank entry 1GRN) via InterPReTS (Aloy and Russell, 2002), which assesses the effect of evolutionary changes at the interface structure using empirical pair potentials (Betts et al., 2015).

We modeled the Cdc42–RhoGAP19D 3D complex using Swiss-Model (Waterhouse et al., 2018) and rendered the interaction interface using the PyMOL Molecular Graphics System version 2.3.0 (Schrödinger, LLC).

### *Drosophila* mutant stocks and transgenic lines
We used the following mutant alleles and transgenic constructs: *cdc42²* (Fehon et al., 1997; Bloomington *Drosophila* Stock Center

[BDSC] 9105), *p120ctn308* (Myster et al., 2003; BDSC 81638), *shgIG29* (BDSC 58471), *lgl⁴* (Gateff, 1978; BDSC 36289), *scrib²* (Bilder and Perrimon, 2000), *fasIIG0336* (Mao and Freeman, 2009; Kyoto Stock Center 111871), *par-1⁶³²³* (Shulman et al., 2000), *aPKCHC* (Chen et al., 2018), *crb⁸F¹⁰⁵* (Tepass et al., 1990), *gekomb1080* (Gontang et al., 2011), *Pak1²²* (Newsome et al., 2000; a gift from the Dickson laboratory, Janelia Research Campus, Ashburn, VA), *CadNΔ14* (Prakash et al., 2005), *α-catenin* RNAi (BDSC 33430), *nrg* RNAi (BDSC 38215), *fasII* RNAi (BDSC 34084), *fasIII* RNAi (BDSC 77396), *scrib* RNAi (BDSC 35748), *cora* RNAi (Vienna *Drosophila* Resource Center 9788), *rhogap19d* RNAi (P{TRiP.HMS00352}attP2; BDSC 32361), E-cadherin–EGFP (BDCS 60584; Huang et al., 2009), Lgl-EGFP (Tian and Deng, 2008), mNeonGreen-NWasp (a gift from Jenny Gallop, Gurdon Institute, Cambridge, UK), mCherryGrabFP-Baz (Harmansa et al., 2017), aPKC-EGFP (Chen et al., 2018), Zipper-EGFP (Lowe et al., 2014), UASp-GFP RhoGAP19D (BDSC 66167), y* w*; P{GawB}NP1624 (Traffic Jam-Gal4; Brand and Perrimon, 1993), and nanos-GAL4 (a gift from Ruth Lehmann, The Whitehead Institute, Massachussetts Institute of Technology, Cambridge, MA). The following stocks were used to generate mitotic clones: ubiRFP-nls, hsflp, FRT20A4 (PBac{WH}f01417; Exelixis), FRT40A ubiRFP-nls (BDSC 34500), FRT82B, ubiRFP-nls (BDSC 30555), and FRT82B ubiGFP (BDSC 5188), y w hs-FLP; Act5C>CD2>Gal4, UAS:mRFPnls (Pignoni and Zipursky, 1997; BDSC 30558).

### Generation of endogenously tagged RhoGAP19D and N-WASP
The mNeonGreen tag (Shaner et al., 2013) was fused to the N-terminus of RhoGAP19D by CRISPR-mediated homologous recombination. In vitro synthesized gRNA to a CRISPR target (target sequence 5′-GGTGGCGACTCCGGCAGCGGCGG-3′, CRISPR; located

25,985 bp from the 5′ end of RhoGAP19D) and a plasmid donor containing the ORF of mNeonGreen as well as appropriate homology arms (1.5 kb upstream and downstream) were coinjected into nos-Cas9–expressing embryos (BDSC 54591). Single F0 flies were mated to y w flies and allowed to produce larvae before the parents were analyzed by PCR. Progeny from F0 flies in which a recombination event occurred (as verified by PCR) were crossed and sequenced to confirm correct integration. Several independent mNeonGreen-RhoGAP19D lines were isolated. Recombinants carry the mNeon-Green coding sequence inserted immediately downstream of the endogenous start codon with a short linker (Gly-Ser-Gly-Ser) between the coding sequence of mNeonGreen and the coding sequence of RhoGAP19D. Homozygous flies are viable and healthy.

The left arm of RhoGAP19D was amplified with the following primers: forward: 5′-TACGACTCACTATAGGGCGAATTGGGT ACCGGGCCCCCCCTCTGTTTGGGGGTAATTACATGTGCTT-3′; reverse: 5′-TCCTCCTCGCCCTTGCTCACCATTTTGTTGCGGGAT ACTGTGGT-3′. The right arm of RhoGAP19D was amplified with the following primers: forward: 5′-TGTACAAGGGAAGCGGTT CCTTACAAAACTCGAACGGCGCT; reverse: 5′-GGCGGCCGCTCTA GAACTAGTGGATCCCCCGGGCTGCAGGTGTAAAAACCTTTT TCTGTGGCATTTAACATAGACC-3′. mNeonGreen was amplified with the following primers: forward: 5′-CAGTATCCCGCAACA AAATGGTGAGCAAGGGCGAGGAG-3′; reverse: 5′-GCGCCGTTC GAGTTTTGTAAGGAACCGCTTCCCTTGTACAG-3′.

To discriminate between the signal of mNeonGreen-RhoGAP19D in the germline and in the somatic follicular cells, UAS-RhoGAP19D RNAi was expressed in the germline under the control of nos-Gal4. The knockdown was efficient, which allowed visualization Neo-RhGAP19D expression in the follicular cells only.

mNeonGreen-N-WASP was generated by CRISPR-mediated insertion of mNeonGreen before exon 2 of N-WASP in order to target all N-WASP isoforms as isoform C lacks exon 1. A linker sequence of four serine residues was added such that the fusion protein junction corresponds to LYKSSSSTLN. Two guide RNA sites were chosen that flanked the insertion site, with the PAM motifs being separated by 17 nucleotides. The donor plasmid pTv-[w+] mNeonGreen N-WASP constructed by In-Fusion cloning of PCR generated 5′ and 3′ homology arms, mNeon-Green and HindIII/XhoI cut pTv-[w+] vector. The 940 bp 5′ homology arm was amplified from genomic DNA with the forward primer 5′-GAATCTGCAGCTCGACGGCTGCAGTGTTTC AATTGCCAG-3′ and the reverse primer 5′-CTCACCATCTGA AAGTGGAGCAAGCAGAGATTG-3′. The 1.2 kb 3′′ homology arm was amplified from genomic DNA with the forward primer 5′-TCGAGCTCATCGACACTCAACACTGCAGTGGTGCAGATCTAC AAG-3′ and the reverse primer 5′-TCGAAAGCCGAAGCTCAT TGATGACTTACCGCCACAACAGG-3′. The forward primer has two silent changes in the gRNA target sequence (C to T and C to A at positions 19 and 22) to prevent cleavage of the donor plasmid. It was not necessary to similarly mutate the 5′ homology arm primers as the mNeonGreen insertion falls within the gRNA sequence. mNeonGreen was amplified from pNCS mNeonGreen (Allele Biotech) with primers 5′-CTTTCAGATGGTGAGCAAGGG CGAGGAGGATAAC -3′ and 5′-TGTCGATGAGCTCGACTTGTA CAGCTCGTCCATGCCCATCACATCGG-3′. NWASP gRNA target

sequences were cloned into pCFD4-U6:1_U6:3tandemgRNAs plasmid by In-Fusion cloning. Primers 5′-TCCGGGTGAACTTCG CGGTGTTGAGTGTCTGAAAGGTTTTTAGAGCTAGAAATAGCAA G-3′ and 5′-TTCTAGCTCTAAAACCCGTGGTGCAGATCTACA AGCGACGTTAAATTGAAAATAGGTC-3′ were used to amplify and introduce the two N-WASP gRNAs into BbsI cut pCFD4-U6: 1_U6:3tandemgRNAs. CDF2 nos-Cas9 fly embryos were injected with 250 ng/μl of the donor and the gRNA construct. 15 X F0 male flies were crossed with TM3/TM6 balancer (injected males were pre-screened by PCR with the forward 5′ homologous arm primer and a reverse mNeonGreen primer 5′-CACCATGTCAAA GTCC-3′). 4 X Positive F1 flies (pre-screened as for F0 flies) were then crossed with a TM3/TM6 balancer line. Correct integration of mNeonGreen was confirmed in two lines by PCR and sequencing across the entire donor sequence. PCR was also performed with primers flanking the donor sequence to confirm the size of the integrated fragment.

### Generation of *rhogap19d* mutant flies

We used the error-prone repair of CRISPR/Cas9-induced double-stranded DNA breaks by the nonhomologous end-joining repair pathway to generate null alleles of *rhogap19d* (Bassett et al., 2013). The following target sequence was used 5′-GGGTCG GGATCCCTTTCGGGGGG-3′, CRISPR (38,613 bp from the 5′ end). In vitro synthesized gRNA to the target sequence and Cas9 mRNA were injected into FRT20A4 embryos. Single F0 flies were mated to y w flies. DNA from F0 flies was extracted and analyzed by high-resolution melting. Short fragments of 137 bp covering the region containing the target sequence were amplified by PCR (primers used were forward: 5′-CACAATCAGGCG TTGTATGC-3′, reverse: 5′-CTCCTCCTTCTGCTTGATGG-3′). Slow melting curves were generated for the PCR products, and changes in sequence were measured by changes in fluorescence as the strands separated. This technique allows the detection of single-base changes. Progeny of promising F0 candidates were balanced and sequenced. Multiple mutant alleles of *rhogap19d* on the FRT20A4 chromosome were isolated and analyzed. The mutants that contain insertions or deletions generating premature STOP codons were kept for clonal analysis. These mutations generate proteins of ~436 aa that lack the RhoGAP domain but contain the PDZ domain. *rhogap19d* mutant flies are semilethal. The follicular cell phenotypes of the *rhogap19d* alleles used in this study are rescued by the expression of UAS-RhoGAP19D-GFP under the control of TJ-Gal4, confirming that they are caused by loss of RhoGAP19D function.

### Reagents

The following primary antibodies were used: anti-Armadillo (N2 7A1 1:100 dilution; Developmental Studies Hybridoma Bank), anti-Dlg (4F3, 1:100 dilution; Developmental Studies Hybridoma Bank), anti-aPKC (C-20, sc-216-G, goat polyclonal IgG, 1:500; Santa Cruz Biotechnology), anti-Cno (1:1,000 dilution; Takahashi et al., 1998; a gift from M. Peifer, University of North Carolina, Chapel Hill, NC) anti-Gek (1:25 dilution; Gontang et al., 2011; a gift from the Clandinin laboratory, Stanford University, Stanford, CA), and anti-PH3 (9701S, 1:500 dilution; Cell Signaling Technology).

The following secondary antibodies were used: Alexa Fluor secondary antibodies (Invitrogen) at a dilution of 1:1,000, Alexa Fluor 488 goat anti-mouse (A11029), Alexa Fluor 488 goat anti-rabbit (A11034), Alexa Fluor 647 goat anti-mouse (A21236), Alexa Fluor 647 goat anti-rabbit (A21245).

F-actin was stained with phalloidin conjugated to rhodamine (R415, 1:500 dilution; Invitrogen). The cell membranes were labeled with CellMask Orange Plasma Membrane Stain or CellMask Deep Red Plasma Membrane Stain (Thermo Fisher Scientific).

### Immunostaining
Ovaries from fattened adult females, salivary glands from L3 instar larvae, gut from L3 instar larvae, and accessory glands from virgin or mated males were dissected in PBS and fixed with rotation for 20 min in 4% paraformaldehyde and 0.2% Tween 20 in PBS. After a few washes with PBS with 0.2% Tween, tissues were then incubated in 10% BSA in PBS to block for at least 1 h at room temperature. Incubations with primary antibodies were performed at 4°C overnight in PBS, 0.2% Tween 20, and 1% BSA. This step was followed by four washes with PBS with 0.2% Tween, and samples were then incubated for 3–4 h with secondary antibody at room temperature. Specimens were then washed several times in washing buffer and mounted in Vectashield containing DAPI (Vector Laboratories).

Embryos were fixed using the formaldehyde/heptane fixation method, followed by methanol extraction.

For staining with the anti-Gek antibody, ovaries were heat fixed as described by Chen et al. (2018).

### Imaging
Fixed samples and live imaging were performed using an Olympus IX81 (40×/1.3 UPlan FLN oil objective or 60×/1.35 UPlanSApo oil objective) or a Leica SP8 white laser (63×/1.4 HC plan apochromat confocal scanning oil objective) inverted confocal microscope. For live observations, ovaries were dissected and imaged in 10S Voltalef oil (VWR Chemicals) at room temperature. Images were processed with Fiji (Schindelin et al., 2012) or Leica analysis software.

### *Drosophila* genetics
Standard procedures were used for *Drosophila* maintenance and experiments. Flies were grown on standard fly food supplemented with live yeast at 25°C. Follicular cell clones were induced by incubating larvae or pupae at 37°C for 2 h every 12 h over a period of at least 3 d. Adult females were dissected at least 2 d after the last heat shock. In some experiments, adult flies were heat shocked for at least 3 d and dissected 1 d after the last heat shock.

We used the Flipout technique with Actin5c>Cd2>Gal4 to generate marked clones of cells expressing RNAi constructs (Fig. S3). Flp recombination was induced by incubating larvae or pupae at 37°C for 2 h every 12 h over a period of at least 3 d.

### Genetic interactions
To test for genetic interactions between *rhogap19d* and adhesion molecules or polarity factors, we analyzed the frequency of follicle cell invasions at stages 7 and 8 in large anterior *rhogap19d*

clones that covered at least 25% of the follicular epithelium in each genetic background.

An unpaired, two-tailed Student's $t$ test with Welch's correction was used to determine whether any differences between the penetrance of the invasion phenotype in *rhogap19d* alone and in combination with each mutant or RNAi knockdown were significant.

### Quantifications of the total number of follicle cells per egg chamber in *rhogap19d* mutants
Confocal z-stacks of whole egg chambers were collected on a Leica SP8 white laser microscope. Each egg chamber was divided in three regions. Nuclei were counted twice per region.

### Reproducibility of experiments
All experiments were repeated multiple times as listed below. For each figure, the first number indicates the number of times that the experiment was repeated, and the second indicates the number of egg chambers or clones analyzed. The number of independent experiments performed were as follows: Fig. 1 A (3, 28); Fig. 2 C (5, 89); Fig. 2 E (3, 34); Fig. 2 F (2, 20); Fig. 2 H (7, 56); Fig. 2 I (4, 34); Fig. 2, J and K (3, 6); Fig. 2 L (2, 9); Fig. 2 M (3, 12); Fig. 2 N (2, 7); Fig. 3 A (4, 24); Fig. 3 B (3, 46); Fig. 3, D and E (3, 67, and 45); Fig. 4 A (7, 67); Fig. 4 B (3, 78); Fig. 4 C (5, 98); Fig. 4 D (2, 16); Fig. 4 E (3, 56); Fig. 5 A (7, 56); Fig. 5 C (5, 47); Fig. 5 D (3, 98); Fig. 5 E (3, 18); and Fig. 6: *rhogap19d* (8, 301), *scrib/+* (3, 194), *lgl/+* (3, 94), *fas2* (3, 185), *nrg* (3, 114), *par1/+* (3, 155), *aPKC/+* (4, 98), *crb* (2, 129), *gek* (2, 78), *Pak1* (3, 228).

### Online supplemental material
Fig. S1 shows that RhoGAP92B, RhoGAP68F, CdGAPr, RacGAP84C, Conu, and RhoGAP93B are not required for follicle cell polarity. Fig. S2 shows that RhoGAP19D localizes laterally in multiple epithelia. Fig. S3 shows that loss of RhoGAP19D causes apical domain expansion in several epithelia. Fig. S4 shows that the *rhogap19d* phenotype resembles the early steps in breast cancer. Video 1 is a time-lapse movie of a stage 7 egg chamber containing a large *rhogap19d* mutant clone (marked by the loss of RFP) and expressing GFP-aPKC. Table S1 lists CRISPR-mediated mutations in candidate Cdc42 GAPS.

## Acknowledgments
We thank Franck Pichaud, Barry Dickson, Marc Peifer, Thomas Clandinin, Ruth Lehmann, Andrea Brand, and their laboratories for fly stocks and for antibodies; the Developmental Studies Hybridoma Bank, the Kyoto Stock Center, the Vienna *Drosophila* Resource Center, Exelixis, the Bloomington *Drosophila* Stock Center, and Santa Cruz Biotechnology for antibodies and fly stocks; Jia Chen for help with heat fixation; Richard Butler for help with image analysis; Nick Lowe for help with high-resolution melting analysis; and members of the D. St Johnston laboratory for technical assistance, helpful comments, and criticism.

W. Fic, R, Bastock, E. Los and D. St Johnston were supported by Wellcome Principal Research fellowships to D. St Johnston (080007 and 207496) and Centre funding from Wellcome

(092096 and 203144) and Cancer Research UK (A14492 and A24843). R.B. Russell acknowledges support from the German Network of Bioinformatics Infrastructure funded by the Bundesministerium für Bildung und Forschung. F. Raimondi was supported by an Alexander Von Humboldt Foundation postdoctoral fellowship.

The authors declare no competing financial interests.

Author contributions: The project was conceived by W. Fic, R. Bastock, R.B. Russell, and D. St Johnston. The structural modeling of *Drosophila* RhoGAP proteins was performed by F. Raimondi and R.B. Russell. The *Drosophila* experiments were performed by W. Fic, R. Bastock, and E. Los, and the data were analyzed by W. Fic, R. Bastock, and D. St Johnston. The mNeonGreenWASP line was generated by Y. Inoue and J. Gallop. The project funding, administration, and supervision were provided by D. St Johnston and R.B. Russell. W. Fic and D. St Johnston prepared the figures and wrote the manuscript, which was edited and reviewed by all authors.

Submitted: 16 September 2020

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

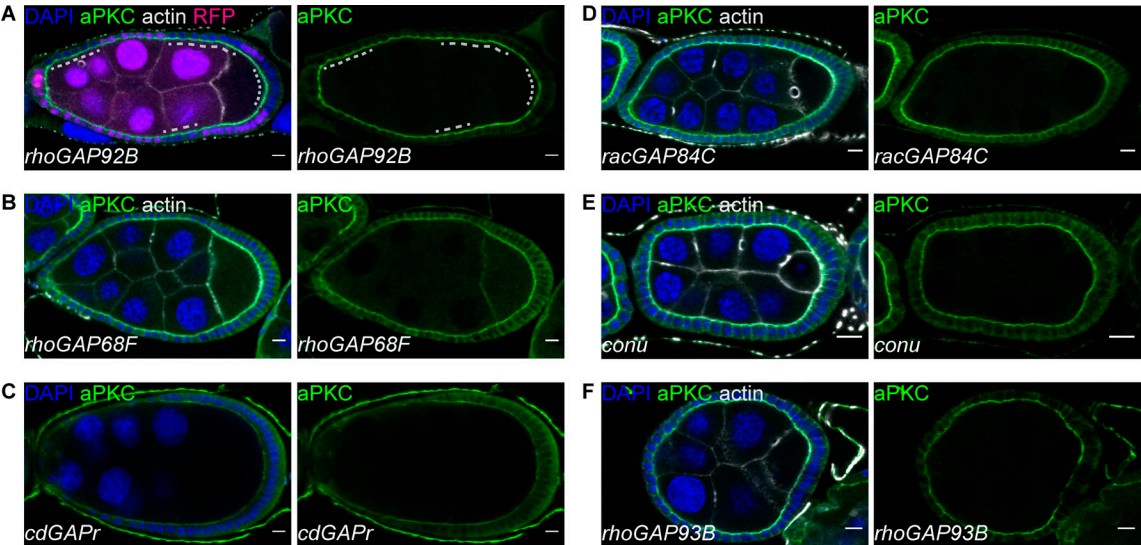

Figure S1. **RhoGAP92B, RhoGAP68F, CdGAPr, RacGAP84C, Conu, and RhoGAP93B are not required for follicle cell polarity. (A)** Egg chambers containing *rhoGAP92B* mutant cells marked by loss of RFP. Loss of RhoGAP92B does not affect aPKC localization (A; *n* = 28) or Lgl localization (not shown; *n* = 24). Dashed lines indicate RFP-negative mutant cells. **(B–F)** Egg chambers from flies homozygous for *rhoGAP68F* (B; *n* = 109), *cdGAPr* (C; *n* = 12), *racGAP84C* (D; *n* = 104), *conu* (E; *n* = 90), and *rhoGAP93* (F; *n* = 119) mutants show normal localization of aPKC (B–F) and Lgl (not shown) and normal organization of the follicular epithelium. Scale bars, 10 µm.

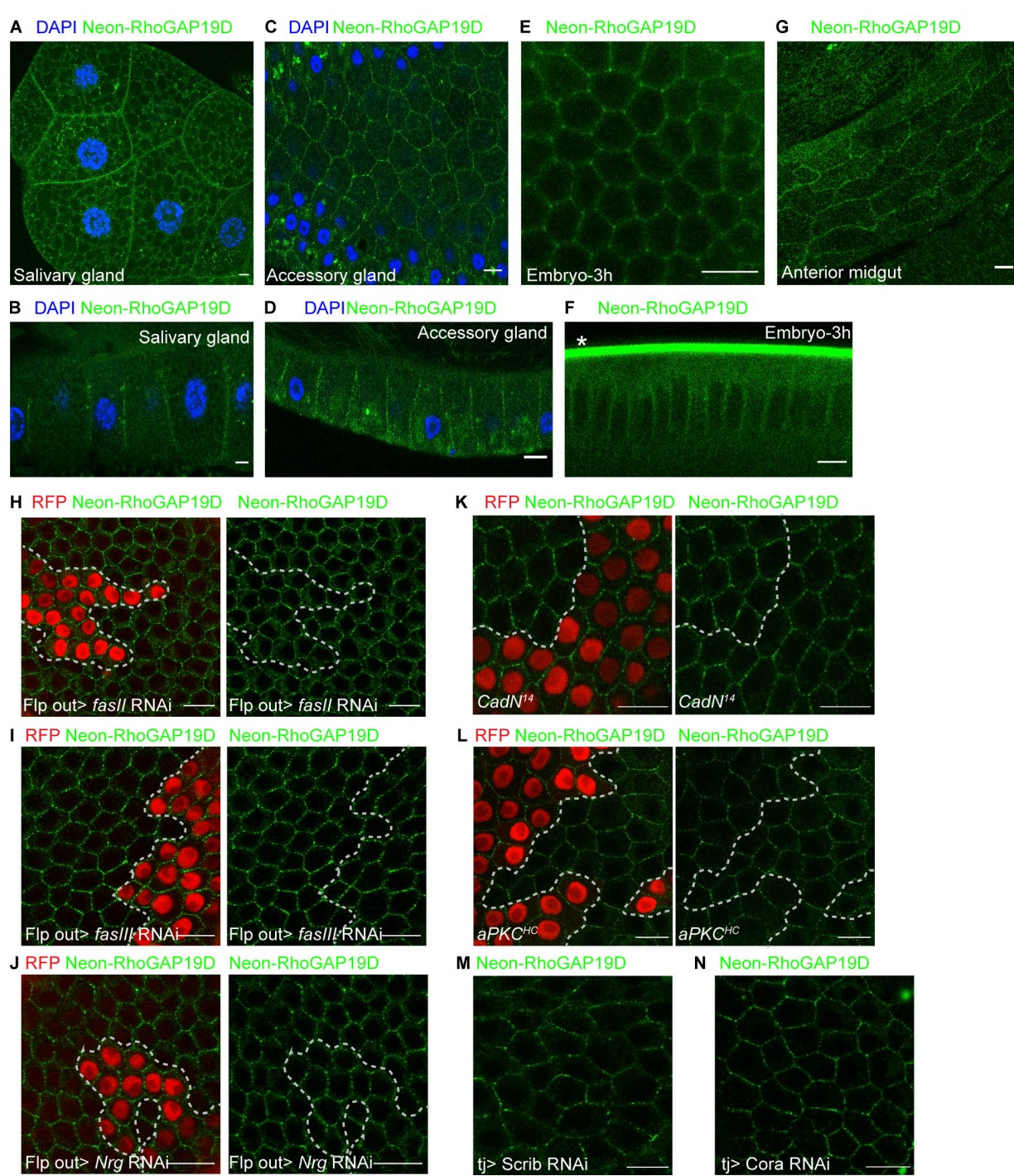

Figure S2. **RhoGAP localizes laterally in multiple epithelia. (A and B)** A top view (A) and a cross-section (B) through a salivary gland from an L3 larva, showing mNeonGreen-RhoGAP19D (green) localization to the lateral domain. DAPI (blue), $n = 8$ larvae. **(C and D)** A top view (C) and a cross-section (D) through an accessory gland from a mated male. mNeonGreen-RhoGAP19D (green) localizes laterally. DAPI (blue), $n = 10$ males. **(E and F)** A top view (E) and a cross-section (F) through a live 3-h-old embryo. mNenonGreen-RhoGAP19D (green) localizes laterally, *Autofluorescence of the vitelline membrane. $n = 11$ embryos. **(G)** Lateral mNeonGreen-RhoGAP19D (green) localization in the anterior midgut of an L3 larva; $n = 3$ guts. **(H–J)** Wild-type localization of mNeonGreen-RhoGAP19D at the lateral cortex of stage 7 follicle cells in which fasII (H; $n = 21$), fasIII (I; $n = 19$), and nrg (J; $n = 25$) have been knocked down using the UAS-RNAi Flp out system (cells that express RFP coexpress the RNAi constructs). Each experiment was performed three times. **(K)** A stage 8 egg chamber with a clone of cells mutant for *N-cadherin1* and *N-cadherin2* (marked by the loss of RFP). mNeonGreen-RhoGAP19D localizes normally in the mutant cells ($n = 15$). The experiment was performed twice. **(L)** mNeonGreen-RhoGAP19D localizes to the cortex in $aPKC^{HC}$ mutant clones (marked by the loss of RFP) at stage 8. $n = 16$; experiment performed twice. **(M and N)** mNeonGreen-RhoGAP19D is correctly localized in cells treated with RNAi against Scrib ($n = 23$; M) or Coracle (N; $n = 16$). Stage 8; experiments performed twice. Scale bars, 10 μm.

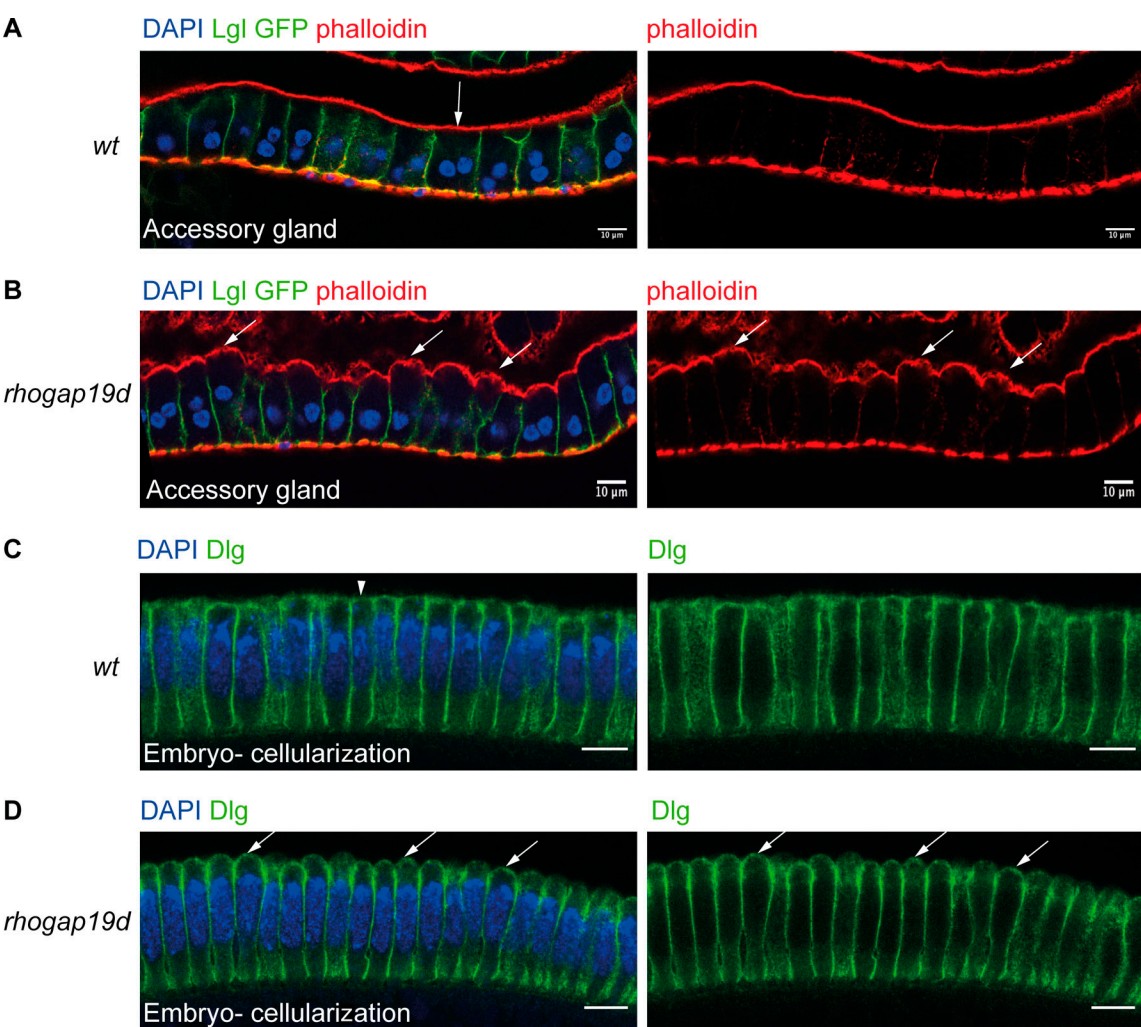

Figure S3.  ***rhogap19d* mutant cells expand apically in different epithelia. (A and B)** Accessory glands from virgin wild-type (A) and *rhogap19d* mutant males (A) stained for Lgl (green), F-actin (phalloidin; red), and DNA (DAPI; blue). The apical surfaces of the mutant cells protrude into domes, and the cells are much taller than in wild type (*n* = 11 males). Scale bars, 10 µm. **(C and D)** Cellular blastoderm embryos (3 h after fertilization) from wild-type (C) and *rhogap19d* mutant mothers, stained for Dlg (green) and DNA (DAPI; blue). The cells in the embryos laid by homozygous mutant females bulge apically. The vertical arrow and arrowhead in A and C indicate the smooth apical surface in wild type, and the diagonal arrows mark the domed apical surfaces in *rhogap19d* mutant cells. *n* = 8 embryos in each background. Scale bars, 10 µm.

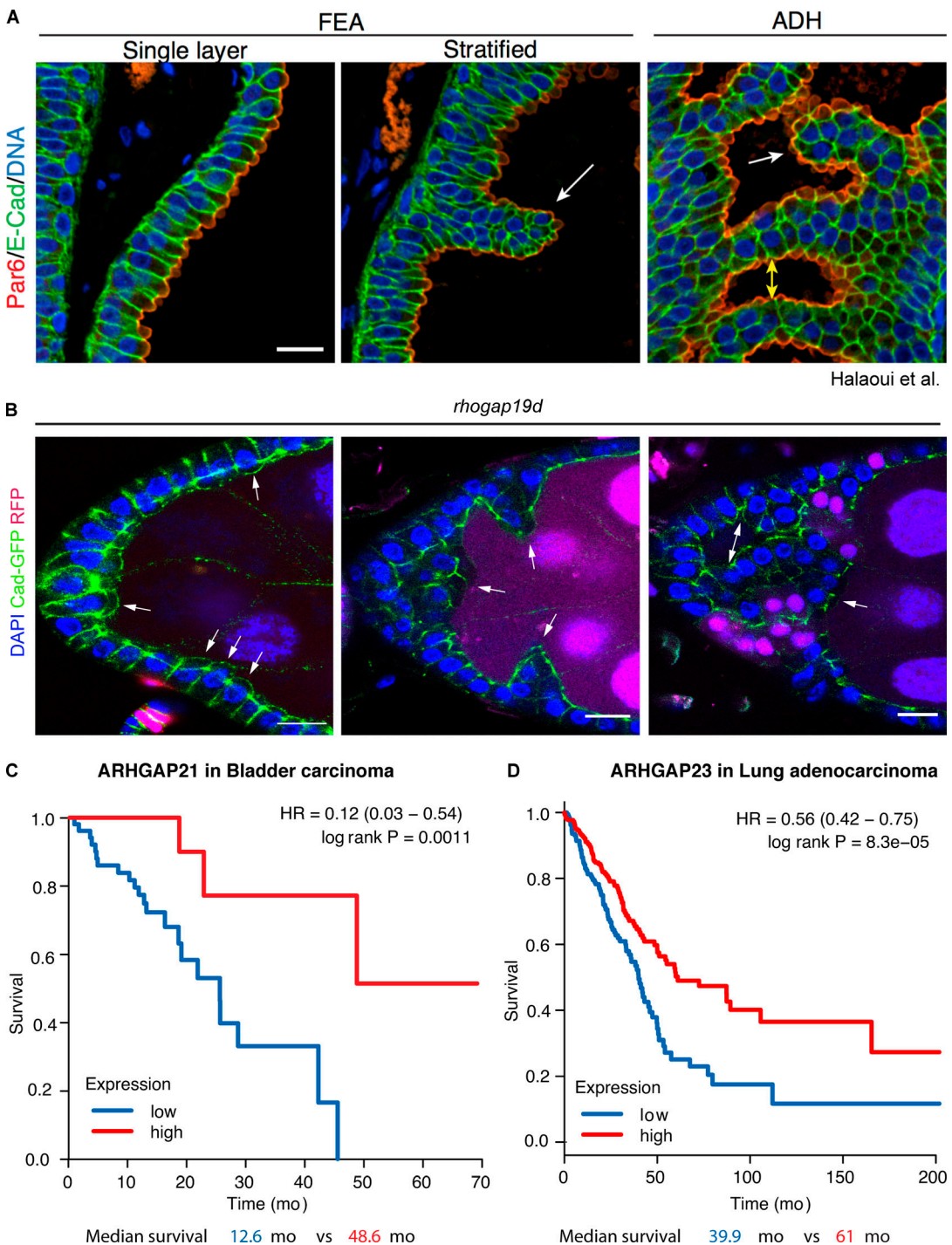

Figure S4.  **The rhogap19d phenotype resembles the early steps in breast cancer. (A)** Images of breast tissue samples reproduced from Halaoui et al. (2017; Fig. S3), reprinted with permission from *Genes & Development*, showing examples of FEA and ADH. Samples were immunostained for Par-6 (red), E-cadherin (E-Cad; green), and DAPI (blue). The white arrows show polarized cells invading into the lumen. The yellow arrows show epithelial bridges that split the primary lumen. **(B)** *rhogap19d* mutant cells (marked by the absence of RFP), stained for E-cadherin–GFP (Cad-GFP; green) and DAPI (blue), show similar apical bulges and invasions to FEA and ADH. Cells first bulge apically (white arrows in the left panel), then start to collectively invade the germline (white arrows in the middle panel), to finally form big clusters inside the egg chamber (white arrows in the right panel). Stage 8 egg chamber. Experiment repeated five times. Scale bars, 10 μm. **(C)** Kaplan-Meier survival plot for ARHGAP21 expression (high; top quartile versus low; bottom quartile) in bladder carcinoma. Survival data were retrieved from the kmplot resource (kmplot.com) described in Györffy et al. (2010). **(D)** Kaplan-Meier survival plot for ARHGAP23 expression in lung adenocarcinoma. HR, hazard ratio, with 95% confidence limits in parentheses.

Video 1.   **A time-lapse movie of a stage 7 egg chamber containing a large *rhogap19d* mutant clone (marked by the loss of RFP; magenta) and expressing GFP-aPKC (green).** Frames were captured every 15 s. Elapsed time, 11 min; playback time, 3 s.

**Table S1 is provided online and lists CRISPR-mediated mutations in candidate Cdc42 GAPs.**

