## [Peer Review File · The Journal of Cell Biology]

RhoGAP19D inhibits Cdc42 laterally to control epithelial cell shape and prevent invasion

Weronika Fic, Rebecca Bastock, Francesco Raimondi, Erinn Los, Yoshiko Inoue, Jennifer Gallop, Robert Russell, and Daniel St Johnston

Corresponding Author(s): Daniel St Johnston, University of Cambridge

Review Timeline:

Submission Date:	2020-09-16
Editorial Decision:	2020-10-14
Revision Received:	2020-12-04
Editorial Decision:	2021-01-04
Revision Received:	2021-01-12

Monitoring Editor: Mark Peifer

Scientific Editor: Andrea Marat

Transaction Report:

DOI: <https://doi.org/10.1083/jcb.202009116>

October 14, 2020

Re: JCB manuscript #202009116

Dr. Daniel St Johnston
University of Cambridge
The Gurdon Institute
Tennis Court Rd
Cambridge CB2 1QN
United Kingdom

Dear Daniel,

Thank you for submitting your manuscript entitled "RhoGAP19D inhibits Cdc42 laterally to control epithelial cell shape and prevent invasion". The manuscript was assessed by expert reviewers, whose comments are appended to this letter. We invite you to submit a revision if you can address the reviewers' key concerns, as outlined here.

As you will see, both reviewers thought the work was carefully done and of potentially broad interest. Reviewer 1 suggests some relatively straightforward revisions to the text and clarifications with regard to quantification. Reviewer 2 suggests some relatively modest extensions to the data presented and some additional quantification. These seem like reasonable issues you should address in your revision.

GENERAL GUIDELINES:

Text limits: Character count for an Article is < 40,000, not including spaces. Count includes title page, abstract, introduction, results, discussion, acknowledgments, and figure legends. Count does not include materials and methods, references, tables, or supplemental legends.

Figures: Articles may have up to 10 main text figures. Figures must be prepared according to the policies outlined in our Instructions to Authors, under Data Presentation, <https://jcb.rupress.org/site/misc/ifora.xhtml>. All figures in accepted manuscripts will be screened prior to publication.

Supplemental information: There are strict limits on the allowable amount of supplemental data. Articles may have up to 5 supplemental figures. Up to 10 supplemental videos or flash animations are allowed. A summary of all supplemental material should appear at the end of the Materials and

methods section.

As you may know, the typical timeframe for revisions is three to four months. However, we at JCB realize that the implementation of social distancing and shelter in place measures that limit spread of COVID-19 also pose challenges to scientific researchers. Lab closures especially are preventing scientists from conducting experiments to further their research. Therefore, JCB has waived the revision time limit. We recommend that you reach out to the editors once your lab has reopened to decide on an appropriate time frame for resubmission. Please note that papers are generally considered through only one revision cycle, so any revised manuscript will likely be either accepted or rejected.

Thank you for this interesting contribution to Journal of Cell Biology. You can contact us at the journal office with any questions, cellbio@rockefeller.edu or call (212) 327-8588.

Sincerely,

Mark Peifer
Monitoring Editor

Andrea L. Marat
Senior Scientific Editor

Journal of Cell Biology

Reviewer #1 (Comments to the Authors (Required)):

Using the *Drosophila* egg chamber follicle as a model system, Fic and colleagues discover a role of RhoGAP19D in controlling the molecular composition of the basolateral domain, cell shape, and tissue organization. A bioinformatics analysis of *Drosophila* Rho family GAPs aimed at identifying Cdc42-specific GAPs selected RhoGAP19D as the top hit. Moreover, analyses of loss-of-function mutants of the top seven hits only revealed follicle structure abnormalities for RhoGAP19D mutants. In RhoGAP19D mutants, abnormal folding of the follicular epithelium was associated with invasion into space normally occupied by germline nurse cells. Endogenously expressed RhoGAP19D localized to the basolateral domain of follicle cells, and this localization required DE-cad and alpha-catenin, but not tested basolateral polarity proteins. The abnormal follicle structure of RhoGAP19D mutants occurred without a change to cell proliferation, but was accompanied by altered cell shapes (bulging apical domains), elevated basolateral localization of the Cdc42 effectors Wasp and MRCK, and elevated basolateral detection of an MRCK phosphorylation target, myosin. With some exceptions, mutants of basolateral factors enhance the RhoGAP19D effect, whereas mutants of apical factors suppress the effect. Comparisons are made between the epithelial defects of RhoGAP19D mutants and of the early stages of breast and bladder cancers, and low expression levels of RhoGAP19D homologs are shown be correlated with reduced survival

with these cancers. Overall, this comprehensive study reports significant findings that should of interest to readers of JCB. A number of points should be addressed.

1. The authors say "RhoGAP19D functions to inhibit Cdc42 laterally and is the only GAP that fulfills this role in the follicle cells", but I don't think their data exclude the possibility of other GAPs also contributing. Mutants of other candidate Cdc42 GAPs lack an epithelial structural defect, but these or other GAPs may still have mild effects on lateral Cdc42. The text could be revised to address this concern. Similarly, the abstract says "[we] identified RhoGAP19D as the sole Cdc42GAP required for polarity in the follicle cell epithelium" but a bioinformatics analysis cannot definitively identify all Cdc42 GAPs, and the authors did not test mutants for all GAPs with reasonable scores.
2. Two explanations are offered for the distinctive genetic interaction between Par-1 and RhoGAP19D (that Par-1 may negatively regulate the other basolateral factors tested, or that Par-1 may positively regulate apical factors), but it was unclear why these possibilities were considered over Par-1 promoting basolateral actomyosin networks.
3. This line was unclear: "The lateral contractility may also contribute to compressive stress because reducing cell height while maintaining a constant volume will exert a pushing force on the neighbouring cells." Total cell height was increased (not reduced) in the mutants. Perhaps the authors mean the reduced height of the lateral domain, but even so, it is unclear how pushing forces would be directed towards neighboring cells versus the apical domain.
4. Could the abnormal enlargement of the apical domain be due, at least in part, to a blebbing-like effect in which contraction of the basolateral domain displaces cytoplasm into the apical domain?
5. In Fig 1C, the human ARHGAP1 sequence should be shown in addition to the Drosophila proteins it is compared to.
6. It should be specified what n values represent (individual cells or follicles) (e.g. Figs 2 and 4). If they represent individual cells, then the numbers of follicles should also be indicated, and statistical tests should be applied to populations of follicles (since cells within a follicle are not independent of each other).
7. n values should be provided for Fig 6.
8. The disruption of Wasp localization in RhoGAP19D clones was somewhat unclear because only a small number of control cells are shown and for only half of these cells is the basolateral domain fully shown. Also, on the right side of the image (Fig 3A) mutant cells display a less abnormal distribution. The image is from one of the poles of the egg chamber, whereas most other data show the side of the chamber.
9. The penetrance of the Wasp, Zipper, and Gek mis-localization should be stated (across egg chambers analyzed).

Reviewer #2 (Comments to the Authors (Required)):

In this manuscript Fic et al. study how active Cdc42 is restricted to the apical domain of epithelial cells, using the follicle epithelium in the fly ovary as a model epithelium. They identify, by generating

multiple new mutant LOF alleles, that out of all *Drosophila* RhoGAPs only the RhoGAP19D affects Cdc42 and prevents its lateral localisation. RhoGAP19D is recruited to adherens junctions by alpha-catenin and its loss leads to cell shape phenotypes due to apical domain expansion and increased lateral myosin activity, thereby inducing bending and buckling of the FC epithelium. Thus, the authors posit, RhoGAP19E at adherens junctions maintains apical active Cdc42, thereby preventing an epithelial layer from folding when this is not appropriate.

This is a very nice and mostly clear-cut study that adds an important link in our understanding of how epithelial polarity is generated and maintained. It will thus be of great interest to a wide audience.

I have a number of comments though that I feel should be addressed:

1) The *rhogap19d* mutant flies are viable, and the authors show expression in salivary glands and accessory glands in the testes, and show a similar apical bulging phenotype in the accessory glands. Where else is RhoGAP19D expressed and are there other tissues that show phenotypes? If not, is this due to complementation by other RhoGAPs?

With regards to the data on accessory glands, in the middle of page 8, the reference to Fig. S3 for accessory gland phenotypes: this should refer to Fig. S2!

Also, the authors show that the *rhogap19d* mutant cells in the accessory glands show apical bulges, but do they also extrude/bend towards the apical side, or is this ovary specific?

2) I find the invasion and buckling terminology slightly misleading or unclear. The authors show that there is no excess proliferation, so any 'compression' or 'buckling' stress cannot arise from this. They show a cell shape change, i.e. enlarged apical domain at the expense of the (baso)lateral domain, which if it were coupled with a reduction of the extent of the basal length of the cell could lead to an apical-out bending of the epithelium (i.e. like an inversion of the 'apical-constriction driven cell wedging'). But this would be due to the cell shape change, not any buckling tension similar to the cited reference where it is proliferation and/or confinement that adds the stress. And I assume cell volume is not changing in what the authors observe here (as they do not comment on it). Otherwise an increase in cell volume of mutant cells could add a stress that together with the cell shape changes could drive buckling.

The authors show, using PH3 staining, that they do not observe an increase in mitoses, but images such as in Figure 2C or Figure S4 B (right hand panel) very much look as if there are surplus follicle cells. Have the authors counted numbers of follicle cells at different stages in the mutants to see if there is a larger number in cross-sections or total follicles?

Along this line, with regards to the comparison to the FEA and ADH stages of breast cancer progression shown in Fig. S4A, I assume these do depend on excessive proliferation? Or are these also phenotypes that appear without any increase in cell number? This should be clearly stated and discussed.

3) Bottom of page 6 (of the pdf/ It would have been useful to have page numbers and/or line numbers to be able to point to the right section of the manuscript with regards to the comments!): What recruits RhoGAP19D to the lateral side below adherens junctions? The authors nicely demonstrate that alpha-catenin is key to the adherens junction recruitment, but it seems important to also know what recruits or maintains it further laterally to prevent active Cdc42 association here!

3) Figure 3, co-overexpression of UAS-GrabFP-A and UAS-RhiGAP19D: the authors show the changed cellular morphology and change in aPKC localisation, but where does (active) Cdc42 localise under these conditions? This would be the direct read-out to say that inducing apical

RhoGAP19D localisation affects apical active Cdc42. Is there no appropriately-tagged reporter? Or at least total Cdc542 (which exist tagged with mCherry).

Also, in the same Figure, panels A: could the authors quantify the change in intensity in lateral versus apical membranes? In the example shown, the difference is most striking in the most posterior follicle cells (at the bottom of the image panel), but the cells at the right of the image that are still mutant show much less WaspNeon laterally. So quantification of several clones from several egg chambers would help to strengthen this point.

4) Reference to Figure 2G at the bottom of page 5 ('...indicating that they....are still epithelial in nature'): Fig.2G does not show this and is the wrong reference.

5) The Gek staining in control and rhogap19d mutant FCs seems to show different stages, the wild-type cells shown must be quite a young egg chamber? Even with cells elongated in the mutant, the cells in F do not quite look like equivalent cells, could this be changed?

6) There are a number of missing small words throughout the manuscript, it would be good to proof read carefully.

Response to the Referees' comments

We are grateful to the referees for their careful consideration of our manuscript. We have taken their reviews into account in preparing a revised version, which we believe is substantially improved. Our responses to specific comments are listed below:

Reviewer 1 Comments for the Author:

Overall, this comprehensive study reports significant findings that should of interest to readers of JCB. A number of points should be addressed.

1. The authors say "RhoGAP19D functions to inhibit Cdc42 laterally and is the only GAP that fulfills this role in the follicle cells", but I don't think their data exclude the possibility of other GAPs also contributing. Mutants of other candidate Cdc42 GAPs lack an epithelial structural defect, but these or other GAPs may still have mild effects on lateral Cdc42. The text could be revised to address this concern. Similarly, the abstract says "[we] identified RhoGAP19D as the sole Cdc42GAP required for polarity in the follicle cell epithelium" but a bioinformatics analysis cannot definitively identify all Cdc42 GAPs, and the authors did not test mutants for all GAPs with reasonable scores.

Our reason for making the statement that RhoGAP19D is the only GAP that functions laterally to inhibit Cdc42 is that we would not see lateral Cdc42 activity if there were a second lateral GAP that repressed it. Nevertheless, we agree with the referee that there might be another GAP that only partially inhibits Cdc42 laterally and we have therefore modified both the abstract and the main text as follows:

"We used sequence analysis and 3D structure modelling to determine which Drosophila GTPase Activating Proteins (GAPs) are likely to interact with Cdc42 and identified RhoGAP19D as the only high probability Cdc42GAP required for polarity in the follicle cell epithelium."

"This implies that RhoGAP19D is the major Cdc42GAP that represses Cdc42 laterally, as no other GAPs can compensate for its loss."

"

2. Two explanations are offered for the distinctive genetic interaction between Par-1 and RhoGAP19D (that Par-1 may negatively regulate the other basolateral factors tested, or that Par-1 may positively regulate apical factors), but it was unclear why these possibilities were considered over Par-1 promoting basolateral actomyosin networks.

We agree with the referee and have amended the text as follows:

"It is also possible that Par-1 acts through the actin cytoskeleton and is required for the lateral contractility induced by ectopic Gek activity."

3. This line was unclear: "The lateral contractility may also contribute to compressive stress because reducing cell height while maintaining a constant volume will exert a pushing force on the neighbouring cells." Total cell height was increased (not reduced) in the mutants. Perhaps the authors mean the reduced height of the lateral domain, but even so, it is unclear how pushing forces would be directed towards neighboring cells versus the apical domain.

We agree with this point, which prompted us to analyse how the loss of RhoGAP19D changes cell shape in more detail. This revealed that the width of mutant cells is 15% less than that of wild-type cells. Since their height increases by 22%, their overall volume is slightly decreased. This argues against strong compressive forces that can promote buckling. However, the mutant cells also undergo pulses of lateral contraction, which will transiently decrease their height and increase their widths, and this could locally increase compression to cause the rare buckling events that initiate invasion. We have amended the text to try to make this clearer:

“Although loss of RhoGAP19D only leads to a partial disruption of polarity, it causes the follicular epithelium to invade the adjacent germline tissue with 40% penetrance. This invasive behaviour is not driven by an epithelial to mesenchymal transition, as the cells retain their apical adherens junctions and epithelial organisation. Instead, the deformation of the epithelium seems to be driven by the combination of an increase in lateral contractility and an expansion of the apical domain, as reducing the dosage of Gek, which activates myosin II to drive the contractility, significantly reduces the frequency of this phenotype, as does halving the dosage of any of the apical polarity factors. The expansion of the apical domain makes the domain too long for the cells to adopt the lowest energy conformation, giving them a tendency to become wedge-shaped, which could drive the invagination. Alternatively, invasion may be driven by the buckling of the epithelium. Recent work has shown that epithelial monolayers under compressive stress and constrained by a rigid external scaffold have a tendency to buckle inwards (Trushko et al., 2020). The follicle cell layer is surrounded by an extracellular matrix that constrains the shape of the egg chamber and which should therefore resist expansion (Haigo and Bilder, 2011). In addition, the pulses of lateral contractility are likely to generate compressive stress, as transiently reducing cell height while maintaining a constant volume will increase the cells’ area, thereby exerting a pushing force on the neighbouring cells. This compression coupled to the tendency to become wedge-shaped due to apical expansion could therefore trigger the rare buckling events that initiate invasion. In support of this view, lateral contractility has been shown to drive the folding of the imaginal wing disc between the prospective hinge region and the pouch (Sui et al., 2018).”

4. Could the abnormal enlargement of the apical domain be due, at least in part, to a blebbing-like effect in which contraction of the basolateral domain displaces cytoplasm into the apical domain?

Since all cells show the apical domes, this cannot be due to the lateral contractions, which are transient and do not occur in all cells at the same time. Furthermore, fixation should relax the contractions, but the apical domes persist. Finally, we see expanded apical domains and doming at the cellular blastoderm stage, where the cells are not yet closed basally. This makes it unlikely that they are due to blebbing caused by increased internal pressure, as the pressure can dissipate basally.

5. In Fig 1C, the human ARHGAP1 sequence should be shown in addition to the Drosophila proteins it is compared to. The ARHGAP1 sequence was shown in the figure, but was labelled with its database code (1grn:B). We have now changed this to ARH1GAP.

6. It should be specified what n values represent (individual cells or follicles) (e.g. Figs 2 and 4). If they represent individual cells, then the numbers of follicles should also be indicated, and statistical tests should be applied to populations of follicles (since cells within a follicle are not independent of each other).

This information was provided in the Reproducibility of experiments section of the Materials and Methods:

“For each figure, the first number indicates the number of times that the experiment was repeated and the second indicates the number of egg chambers or clones analysed. The number of independent experiments performed were: Figure 1a (3; 28), Figure 2c (5; 89), Figure 2e (3; 34), Figure 2f (2; 20), Figure 2h (7; 56), Figure 2i (4; 34), Figure 2j&k (3; 6), Figure 2l (2; 9), Figure 2m (3; 12), Figure 2n (2; 7). Figure 3a (4; 24), Figure 3b (3; 46). Figure 4a (7; 67), Figure 4b (3; 78), Figure 4c (5; 98), Figure 4d (2; 16), Figure 4e (3; 56). Figure 5a (7; 56), Figure 5c (5; 47), Figure 5d (3; 98), Figure 5e (3; 18). Figure 6: rhogap19d (8; 301), scrib/+ (3; 194), lgl/+ (3; 94), fas2 (3; 185), nrg (3; 114), par1/+ (3; 155), aPKC/+ (4; 98), crb (2; 129), gek (2; 78), Pak1 (3; 228).”

7. n values should be provided for Fig 6.

We have now added the number of egg chambers analyzed for each genotype to the Figure, as well as the Reproducibility of Experiments section.

8. The disruption of Wasp localization in RhoGAP19D clones was somewhat unclear because only a small number of control cells are shown and for only half of these cells is the basolateral domain fully shown. Also, on the right side of the image (Fig 3A) mutant cells display a less abnormal distribution. The image is from one of the poles of the egg chamber, whereas most other data show the side of the chamber.

WASP is expressed at very low levels and there is no good antibody to stain fixed samples, so all of these images have to be collected in live egg chambers. The reason we showed an image from the posterior pole of the egg chamber is that the endogenously-tagged mNeon-WASP is expressed at higher levels in these cells than in the lateral cells, making the difference in its localisation between the wild-type and mutant cells easier to see. As we do not have the sensitivity to detect Wasp in apical-basal sections of the follicle cells on the side of the egg chamber, we have now added an extra panel to figure 3 (Fig. 3 B) showing a lateral view through mutant and nonmutant cells on the side of the egg chamber. This shows a clear lateral localisation of Wasp in the mutant cells that is not present in the wild-type cells.

9. The penetrance of the Wasp, Zipper, and Gek mis-localization should be stated (across egg chambers analyzed).

We have now included these data in the Figure legends:

Figure 3 legend

A) The CDC42 effector, N-WASP (tagged with mNeonGreen) spreads laterally in rhogap19D mutant cells (marked by the loss of RFP, magenta). This phenotype was observed in 20/27 mutant cells.

Figure 5 legend

D) rhogap19d mutant cells have lateral foci of non-muscle Myosin II foci (Zipper-GFP; green) and reduced levels at the apical side, compared to wild-type cells. (DAPI; blue). This phenotype was observed in 154/157 mutant cells.

E) Gek (green) localizes apically in wild-type follicle cells but extends along the lateral domain of all rhogap19d mutant cells. DAPI (blue). Scale bars 10µm. N = 11 homozygous mutant egg chambers.

Reviewer #2

1) The rhogap19d mutant flies are viable, and the authors show expression in salivary glands and accessory glands in the testes, and show a similar apical bulging phenotype in the accessory glands. Where else is RhoGAP19D expressed and are there other tissues that show phenotypes? If not, is this due to complementation by other RhoGAPs?

We have now examined RhoGAP19D expression in a number of other epithelia, including the imaginal discs, the cellular blastoderm embryo and in the midgut and observed lateral localisation in each case. We have now added images of the cellular blastoderm embryo and the embryonic midgut as Fig S2 E-G. We have also analysed the effects of loss of maternal RhoGAP19D on the embryo and observe apical bulging in the cellular blastoderm. The embryos eventually die later in embryogenesis or just after hatching, but we have not investigated the phenotype in detail.

rhogap19d mutant flies are only semi-viable, as two thirds of them die before adulthood. The survival of the escapers is probably due to a large maternal contribution, as *rhogap19d* mutants are fully-penetrant maternal effect lethals. We therefore think that it is unlikely that there is a redundant lateral Cdc42GAP in the tissues that we have examined.

We have added this information to the text:

“A similar lateral localisation was observed in all other epithelia we examined, such as the salivary gland, the testis accessory gland, the embryonic midgut and the cellular blastoderm embryo (Fig S2 A-F). Thus, RhoGAP19D seems to be a lateral factor in multiple epithelia. This is consistent with the observation that RhoGAP19d mutants die at several stages. Zygotic RhoGAP19d mutants are semi-lethal, with about two thirds of homozygotes dying before adulthood. Furthermore, all embryos from surviving homozygous mutant mothers either fail to hatch or die as first instar larvae, indicating that it is a fully-penetrant maternal-effect lethal.”

With regards to the data on accessory glands, in the middle of page 8, the reference to Fig. S3 for accessory gland phenotypes: this should refer to Fig. S2! **Done**

Also, the authors show that the rhogap19d mutant cells in the accessory glands show apical bulges, but do they also extrude/bend towards the apical side, or is this ovary specific? We have only observed invasion in the follicle cell epithelium, but have not analysed any other epithelia in detail.

2) I find the invasion and buckling terminology slightly misleading or unclear. The authors

show that there is no excess proliferation, so any 'compression' or 'buckling' stress cannot arise from this. They show a cell shape change, i.e. enlarged apical domain at the expense of the (baso)lateral domain, which if it were coupled with a reduction of the extent of the basal length of the cell could lead to an apical-out bending of the epithelium (i.e. like an inversion of the 'apical-constriction driven cell wedging'). But this would be due to the cell shape change, not any buckling tension similar to the cited reference where it is proliferation and/or confinement that adds the stress. And I assume cell volume is not changing in what the authors observe here (as they do not comment on it). Otherwise an increase in cell volume of mutant cells could add a stress that together with the cell shape changes could drive buckling.

The authors show, using PH3 staining, that they do not observe an increase in mitoses, but images such as in Figure 2C or Figure S4 B (right hand panel) very much look as if there are surplus follicle cells. Have the authors counted numbers of follicle cells at different stages in the mutants to see if there is a larger number in cross-sections or total follicles?

We quantified the total number of follicle cells in wildtype and entirely mutant egg chambers in Fig 2 D. This showed that there are no surplus cells in the latter, ruling out excessive mitosis as a driver for invasion. We agree with the referee that the invasion is likely to be driven by wedging, but consider that buckling may also be involved, given the role of lateral contractility in this process. The cells are slightly narrower than wildtype after fixation, which should reduce compressive stress, but the transient lateral contractions will locally increase compressive stress. We have now revised this section to discuss these alternatives:

“Although loss of RhoGAP19D only leads to a partial disruption of polarity, it causes the follicular epithelium to invade the adjacent germline tissue with 40% penetrance. This invasive behaviour is not driven by an epithelial to mesenchymal transition, as the cells retain their apical adherens junctions and epithelial organisation. Instead, the deformation of the epithelium seems to be driven by the combination of an increase in lateral contractility and an expansion of the apical domain, as reducing the dosage of Gek, which activates myosin II to drive the contractility, significantly reduces the frequency of this phenotype, as does halving the dosage of any of the apical polarity factors. The expansion of the apical domain makes the domain too long for the cells to adopt the lowest energy conformation, giving them a tendency to become wedge-shaped, which could drive the invagination. It is also possible that buckling of the epithelium contributes to invasion. Recent work has shown that epithelial monolayers under compressive stress and constrained by a rigid external scaffold have a tendency to buckle inwards (Trushko et al., 2020). The follicle cell layer is surrounded by an extracellular matrix that constrains the shape of the egg chamber and which should therefore resist expansion (Haigo and Bilder, 2011). In addition, the pulses of lateral contractility are likely to generate compressive stress, as transiently reducing cell height while maintaining a constant volume will increase the cells' cross-sectional area, thereby exerting a pushing force on the neighbouring cells. This compression coupled to the tendency to become wedge-shaped due to apical expansion could therefore trigger the rare buckling events that initiate invasion.”

Along this line, with regards to the comparison to the FEA and ADH stages of breast cancer progression shown in Fig. S4A, I assume these do depend on excessive proliferation? Or are

these also phenotypes that appear without any increase in cell number? This should be clearly stated and discussed.

We have added the following text to address this point:

“In the next stage, atypical ductal hyperplasia (ADH), the ductal cells start to invade the lumen of the duct, while retaining aspects of normal apical-basal polarity (Fig S4). This again resembles the invasive phenotype of rhogap19d mutants, although over-proliferation of the ductal cells probably also contributes to invasion in this case.”

3) Bottom of page 6 (of the pdf/ It would have been useful to have page numbers and/or line numbers to be able to point to the right section of the manuscript with regards to the comments!):

We have now added page numbers.

What recruits RhoGAP19D to the lateral side below adherens junctions? The authors nicely demonstrate that alpha-catenin is key to the adherens junction recruitment, but it seems important to also know what recruits or maintains it further laterally to prevent active Cdc42 association here!

alpha-catenin RNAi removes all lateral RhoGAP19D localisation, not just the localisation to the apical adherens junctions. We therefore presume that the lateral localisation is mediated by binding to alpha-catenin in spot adherens junctions below the main apical adherens junction.

3) Figure 3, co-overexpression of UAS-GrabFP-A and UAS-RhoGAP19D: the authors show the changed cellular morphology and change in aPKC localisation, but where does (active) Cdc42 localise under these conditions? This would be the direct read-out to say that inducing apical RhoGAP19D localisation affects apical active Cdc42. Is there no appropriately-tagged reporter? Or at least total Cdc42 (which exist tagged with mCherry).

Unfortunately, this experiment is not possible because the UAS-GrabFP-A construct is marked with mCherry and the RhoGAP19 with GFP. This rules out using any GFP or RFP tagged constructs to monitor Cdc42 activity and there are no reporters for Cdc42 activity that use other fluorescent proteins.

Also, in the same Figure, panels A: could the authors quantify the change in intensity in lateral versus apical membranes? In the example shown, the difference is most striking in the most posterior follicle cells (at the bottom of the image panel), but the cells at the right of the image that are still mutant show much less WaspNeon laterally. So quantification of several clones from several egg chambers would help to strengthen this point.

As we explained in our response to reviewer 1, the reason we showed an image from the posterior pole of the egg chamber is that the endogenously-tagged mNeon-WASP is expressed at higher levels in these cells than in the lateral cells, making the difference in its localisation between the wild-type and mutant cells easier to see. As we do not have the sensitivity to detect Wasp in apical-basal sections of the follicle cells on the side of the egg chamber, we have now added an extra panel to figure 3 (Fig. 3 B) showing a lateral view through mutant and nonmutant cells on the side of the egg chamber. This shows a clear

lateral localisation of Wasp in the mutant cells that is not present in the wild-type cells. We have also added the quantification of the penetrance of this phenotype, as requested.

4) Reference to Figure 2G at the bottom of page 5 ('...indicating that they.....are still epithelial in nature'): Fig.2G does not show this and is the wrong reference.

Our apologies for this mistake. This is shown in Fig 2E, which is now cited correctly

5) The Gek staining in control and rhogap19d mutant FCs seems to show different stages, the wild-type cells shown must be quite a young egg chamber? Even with cells elongated in the mutant, the cells in F do not quite look like equivalent cells, could this be changed?

We thank the referee for pointing out this, which was due to a mistake in image cropping that meant that the wild-type image was shown at lower magnification. We have now corrected this.

6) There are a number of missing small words throughout the manuscript, it would be good to proof read carefully.

We have done our best to correct these mistakes.

January 4, 2021

RE: JCB Manuscript #202009116R

Dr. Daniel St Johnston
University of Cambridge
The Gurdon Institute
Tennis Court Rd
Cambridge CB2 1QN
United Kingdom

Dear Dr. St Johnston:

Thank you for submitting your revised manuscript entitled "RhoGAP19D inhibits Cdc42 laterally to control epithelial cell shape and prevent invasion". We would be happy to publish your paper in JCB pending final revisions necessary to meet our formatting guidelines (see details below).

A. MANUSCRIPT ORGANIZATION AND FORMATTING:

Full guidelines are available on our Instructions for Authors page, <https://jcb.rupress.org/submission-guidelines#revised>. **Submission of a paper that does not conform to JCB guidelines will delay the acceptance of your manuscript.**

- 1) Text limits: Character count for Articles is < 40,000, not including spaces. Count includes title page, abstract, introduction, results, discussion, acknowledgments, and figure legends. Count does not include materials and methods, references, tables, or supplemental legends.
- 2) Figures limits: Articles may have up to 10 main text figures.
- 3) Figure formatting: Scale bars must be present on all microscopy images, including inset magnifications. Molecular weight or nucleic acid size markers must be included on all gel electrophoresis.
- 4) Statistical analysis: Error bars on graphic representations of numerical data must be clearly described in the figure legend. The number of independent data points (n) represented in a graph must be indicated in the legend. Statistical methods should be explained in full in the materials and methods. For figures presenting pooled data the statistical measure should be defined in the figure legends. Please also be sure to indicate the statistical tests used in each of your experiments (either in the figure legend itself or in a separate methods section) as well as the parameters of the test (for example, if you ran a t-test, please indicate if it was one- or two-sided, etc.). Also, if you used parametric tests, please indicate if the data distribution was tested for normality (and if so, how). If not, you must state something to the effect that "Data distribution was assumed to be normal but this was not formally tested."

5) Abstract and title: The abstract should be no longer than 160 words and should communicate the significance of the paper for a general audience. The title should be less than 100 characters including spaces. Make the title concise but accessible to a general readership.

To increase the accessibility of your title to a broader cell biology audience, we suggest the following:

Spatial control of Cdc42 by RhoGAP19D suppresses epithelial invasion into the adjacent tissue

6) Materials and methods: Should be comprehensive and not simply reference a previous publication for details on how an experiment was performed. Please provide full descriptions in the text for readers who may not have access to referenced manuscripts.

7) Please be sure to provide the sequences for all of your primers/oligos and RNAi constructs in the materials and methods. You must also indicate in the methods the source, species, and catalog numbers (where appropriate) for all of your antibodies. Please also indicate the acquisition and quantification methods for immunoblotting/western blots.

8) Microscope image acquisition: The following information must be provided about the acquisition and processing of images:

a. Make and model of microscope

b. Type, magnification, and numerical aperture of the objective lenses

c. Temperature

d. Imaging medium

e. Fluorochromes

f. Camera make and model

g. Acquisition software

h. Any software used for image processing subsequent to data acquisition. Please include details and types of operations involved (e.g., type of deconvolution, 3D reconstitutions, surface or volume rendering, gamma adjustments, etc.).

10) Supplemental materials: There are strict limits on the allowable amount of supplemental data. Articles may have up to 5 supplemental display items (figures and tables). Please also note that tables, like figures, should be provided as individual, editable files. A summary of all supplemental material should appear at the end of the Materials and methods section.

13) ORCID IDs: ORCID IDs are unique identifiers allowing researchers to create a record of their

various scholarly contributions in a single place. At resubmission of your final files, please consider providing an ORCID ID for as many contributing authors as possible.

B. FINAL FILES:

-- High-resolution figure and video files: See our detailed guidelines for preparing your production-ready images, <https://jcb.rupress.org/fig-vid-guidelines>.

Thank you for this interesting contribution, we look forward to publishing your paper in Journal of Cell Biology.

Sincerely,

Mark Peifer
Monitoring Editor

Andrea L. Marat

Senior Scientific Editor

Journal of Cell Biology

Reviewer #1 (Comments to the Authors (Required)):

The authors have effectively addressed my past concerns. This is an interesting paper that makes an important contribution to the understanding of molecular regulation of epithelial polarity and structure.

Reviewer #2 (Comments to the Authors (Required)):

The authors have done a very good job in addressing both reviewers' concerns. I am happy with the current revised version of the manuscript.

Just one little comment in terms of use of terminology: I would call the folding of the epithelium in this case and evaginations, as it bends out towards apical, in contrast to the usual morphogenetically occurring invaginations, which are bending events towards the basal side.